# Patient-derived colon epithelial organoids reveal lipid-related metabolic dysfunction in pediatric ulcerative colitis

Babajide A. Ojo[1,2,12], Ying Zhu[1,2], Lyong Heo[3], Sejal R. Fox[4], Yanli Qiao[1,2], Amanda Waddell[5,6], Maria E. Moreno-Fernandez [5,7], Marielle Gibson[1,2], Tracy Tran[1,2], Ashley L. Dunn[1,2], Essam I. A. Elknawy[1,2], Neetu Saini[8], Javier A. López-Rivera[1,2], Senad Divanovic [5,6,9], Yuqin Dai[10], Vinicio A. de Jesus Perez [11] & Michael J. Rosen [1,2] ✉

Ulcerative colitis (UC) is associated with epithelial metabolic derangements which exacerbate gut inflammation. Here, we develop colon organoid (colonoid) lines from pediatric patients with endoscopically active UC, inactive UC, and those without intestinal inflammation to interrogate functional metabolic differences in the colon epithelia. We demonstrate that active UC colonoids exhibit hypermetabolic features and cellular stress, specifically during differentiation. Hypermetabolism in active UC colonoids is driven, in part, by increased proton leak, and excess lipid accumulation. Active UC colonoids exhibit heightened activation of the master lipid regulator PPAR-α and its transcriptional pathways. Pharmacological PPAR-α inhibition limits lipid accumulation, induces a metabolic shift towards glucose utilization, suppresses hypermetabolism, and reduces chemokine secretion and cellular stress markers. Collectively, our findings identify lipid-related metabolic dysfunction as a key pathologic feature of the pediatric UC epithelium and highlight the potential of patient-derived colonoids as a preclinical model for evaluating epithelial-targeted therapies addressing this dysfunction.

The restoration of normal intestinal epithelial function is critical for colon mucosa healing in inflammatory bowel diseases (IBD)[1,2]; however, current therapies predominantly target the immune system. As such, only 20–30% of patients achieve mucosal healing with today's therapies[3–6]. Ulcerative colitis (UC) is an IBD subtype characterized by the continuous chronic inflammation of the colon mucosa, extending proximally from the rectum[7]. Pediatric UC is marked by more severe and extensive disease than adult UC[7]. Achieving mucosa healing is one of the best predictors of sustained disease remission[8]. Studies that enhance our understanding of the human intestinal epithelial function

[1]Division of Pediatric Gastroenterology, Hepatology, and Nutrition, Stanford University School of Medicine, Stanford, CA, USA. [2]Center for IBD and Celiac Disease, Stanford Medicine Children's Health, Palo Alto, CA, USA. [3]Stanford Center for Genomics and Personalized Medicine, Stanford University, Palo Alto, CA, USA. [4]Applied Gene and Cell Therapy Center, Cincinnati Children's Hospital Medical Center, Cincinnati, OH, USA. [5]Department of Pediatrics, University of Cincinnati College of Medicine, Cincinnati, OH, USA. [6]Division of Immunobiology, Cincinnati Children's Hospital Medical Center, Cincinnati, OH, USA. [7]Division of Gastroenterology, Hepatology, and Nutrition, Cincinnati Children's Hospital Medical Center, Cincinnati, OH, USA. [8]Division of Pediatric Hematology, Oncology, Stem Cell Transplantation and Regenerative Medicine, Stanford University School of Medicine, Palo Alto, CA, USA. [9]Center for Inflammation and Tolerance, Cincinnati Children's Hospital Medical Center, Cincinnati, OH, USA. [10]Sarafan ChEM-H, Stanford University, Stanford, CA, USA. [11]Division of Pulmonary and Critical Care Medicine, Department of Medicine, Stanford University, Stanford, CA, USA. [12]Present address: Division of Pediatric Gastroenterology, Hepatology, and Nutrition, Heersink School of Medicine, University of Alabama at Birmingham, Birmingham, AL, USA. ✉e-mail: rosenm@stanford.edu

in health and disease may advance epithelial-directed therapies in pediatric UC.

Systemic and local dysregulation of metabolism is a hallmark of inflammation in human IBD[9,10]. Direct investigation of the inflamed human colon tissue in UC has proposed several metabolic impairments in UC, including impaired mitochondrial respiratory chain complex activity, enhanced epithelial mitochondria fission, deficient butyrate metabolism, and reduced epithelial mitochondrial membrane potential[11–14]. This may be critical to the course of disease since mitochondrial activity regulates several aspects of epithelial cellular homeostasis, including the maintenance of energetic balance, adequate reactive oxygen species (ROS) production, and regulating pro-inflammatory responses[15]. However, the cellular mechanisms underlying UC epithelial metabolic dysfunction are not known. Such a deeper understanding may reveal novel epithelial targets to promote mucosal healing.

The development of epithelial-directed therapies in UC is hindered by a lack of human preclinical models of the diseased epithelium. Over the last decade, advances in organoid biology have grown their promise as human pre-clinical models of disease[16]. In particular, patient-derived adult stem cell organoids are capable of self-organizing into structural and functional models that may reflect the cell and molecular complexities of parent tissues[17]. In UC, patient-derived colon organoids (colonoids) may reflect disease-specific disparities compared to healthy colonoids, including alterations in transcription, epigenetics, and somatic gene mutations[18–22]. Despite the promise of intestinal organoids in understanding human intestinal diseases, they have not been used to interrogate the metabolic nature of disease in pediatric UC.

In this work, we use patient-derived colon organoids to investigate epithelial metabolic dysfunction in pediatric ulcerative colitis (UC). We show that hypermetabolism and lipid metabolic dysregulation contribute to epithelial differentiation impairments, cellular stress, and chemokine production in colonoids from pediatric patients with active UC. Furthermore, we identify PPAR-α as a major driver of epithelial metabolic dysfunction in active UC colonoids, and show that inhibition of PPAR-α ameliorates lipid-induced metabolic dysregulation, cellular stress, and inflammation.

## Results

### Derivation of pediatric colon organoids

We developed organoids from rectal biopsies obtained from pediatric patients ($n = 24$) undergoing clinically indicated colonoscopies (Supplementary Fig. 1A, see Supplementary Table 1 for patient demographics). We observed enhanced recovery of matured non-differentiated spheroids from biopsies obtained from non-IBD controls by day 7 (Supplementary Fig. 1B), whereas aUC spheroids required 10–14 days. After the extended growth period, fully established aUC spheroids were morphologically comparable to controls and maintained similarity after multiple passages and a freeze-thaw cycle (Supplementary Fig. 1C, D). The timeline and morphology of spheroids generated from inactive UC (iUC) biopsies (patients with endoscopic healing from active disease) were similar to those of biopsies from non-IBD patients. Overall, we had a 93%, 82%, and 89% success rate in generating stable spheroids from colon biopsies obtained from control, aUC, and iUC patients, respectively.

### Colonoids from active UC patients exhibit impaired differentiation and hypermetabolism

Under non-differentiated conditions, we found no differences in the spheroid diameter between control and aUC spheroids (Supplementary Fig. 2A and B). Accordingly, we observed no metabolic differences between control and aUC spheroids (Supplementary Fig. 2C–F). Similarly, we analyzed extracellular LDH release into the medium – a marker cellular cytotoxicity – and found no differences in LDH release

between control and aUC spheroids (Supplementary Fig. 2G). These results suggest that control and aUC spheroids are morphologically and metabolically similar in their undifferentiated states.

In contrast, we observed morphological differences between aUC and control colon organoids under differentiating conditions (hereafter referred to as colonoids) as early as day 3 as more aUC colonoids remained in their cystic morphology compared to the largely budded structures of control colonoids (Fig. 1A, B). Further differentiation up to 7 days resulted in visibly reduced survival in aUC colonoids compared to controls (Supplementary Fig. 2H). Immunofluorescence staining of day 3 colonoids showed that the goblet cell marker protein MUC2 and colonocyte marker SLC26A3 were reduced in aUC compared to C colonoids, whereas the stem/progenitor marker OLFM4 and tuft cells marker Advillin were similar between the groups. (Supplementary Fig 3A–D). To complement these findings, we used paired spheroids and colonoids (before and after 3-day differentiation) to determine the gene expression of the secretory lineage progenitor, *ATOH1*, the enteroendocrine cell marker, *CHGA*, and the crypt-based goblet cell marker of the colon, *WFDC2*[23], in C and aUC. None of the examined markers were significantly different in control and aUC spheroids (Fig. 1C, and Supplementary Fig. 3E and F). As expected, *ATOH1* and *CHGA* were upregulated in both control and aUC colonoids compared to their respective undifferentiated spheroids, while *CHGA* was lower in aUC compared to C colonoids (Supplementary Fig. 3E and F). In aUC colonoids, there was a failure to upregulate *WFDC2* during differentiation, consistent with the phenomenon of goblet cell depletion in UC (Fig. 1C)[23].

Following the evidence of impaired differentiation in aUC colonoids, we examined whether the metabolic profiles of control and aUC colonoids diverged during the early stages of differentiation. After day 3 of differentiation, we observed a 122% elevation of basal OCR in aUC colonoids compared to the controls (Fig. 1D–F). Since we had more C colonoid lines from female participants than aUC lines, we examined basal OCR between C and aUC colonoids stratified by sex. We continued to observe significantly higher basal OCR in aUC compared to C colonoids within each sex, confirming the effect was not confounded by participant sex (Supplementary Fig. 3G). In addition, we found a 75% significant elevation in median OCR due to proton leak in aUC colonoids compared to control, with a non-significant increase in non-mitochondrial and ATP-linked OCR compared to control colonoids (Fig. 1D–F). Similarly, we found no differences in total cellular ATP between aUC and control colonoids (Fig. 1G). Colon epithelial cells from treatment-naïve UC patients exhibit lower mitochondrial membrane potential (MMP)[14], and we observed a similar reduction in MMP in aUC colonoids compared to controls (Fig. 1H). In line with the hypermetabolic patterns in aUC colonoids, we observed a significant elevation of mitochondrial mass (mtMass) during differentiation compared to controls (Fig. 1I). Correspondingly, we found significant increases in mitochondrial ROS levels and LDH release in aUC colonoids compared to control, suggesting that hypermetabolism in aUC colonoids is accompanied by enhanced oxidative stress and cytotoxicity (Fig. 1J and K). These data suggest hypermetabolism and metabolic inefficiency in aUC colonoids at the early stages of differentiation which is associated with cellular stress.

### Colonoids from inactive UC patients recall features of hypermetabolism upon inflammatory exposure

We asked whether colonoids from inactive UC (iUC) patients were metabolically distinct from control colonoids. First, we observed similar morphological and metabolic differences between control and iUC colonoids during differentiation (Supplementary Fig. 4A–E); however, non-mitochondrial OCR was higher in iUC compared to control colonoids (Supplementary Fig. 4D). Similar to our observation in aUC colonoids, we observed enhanced ROS in iUC colonoids compared to control colonoids. Since biopsies for aUC colonoids were

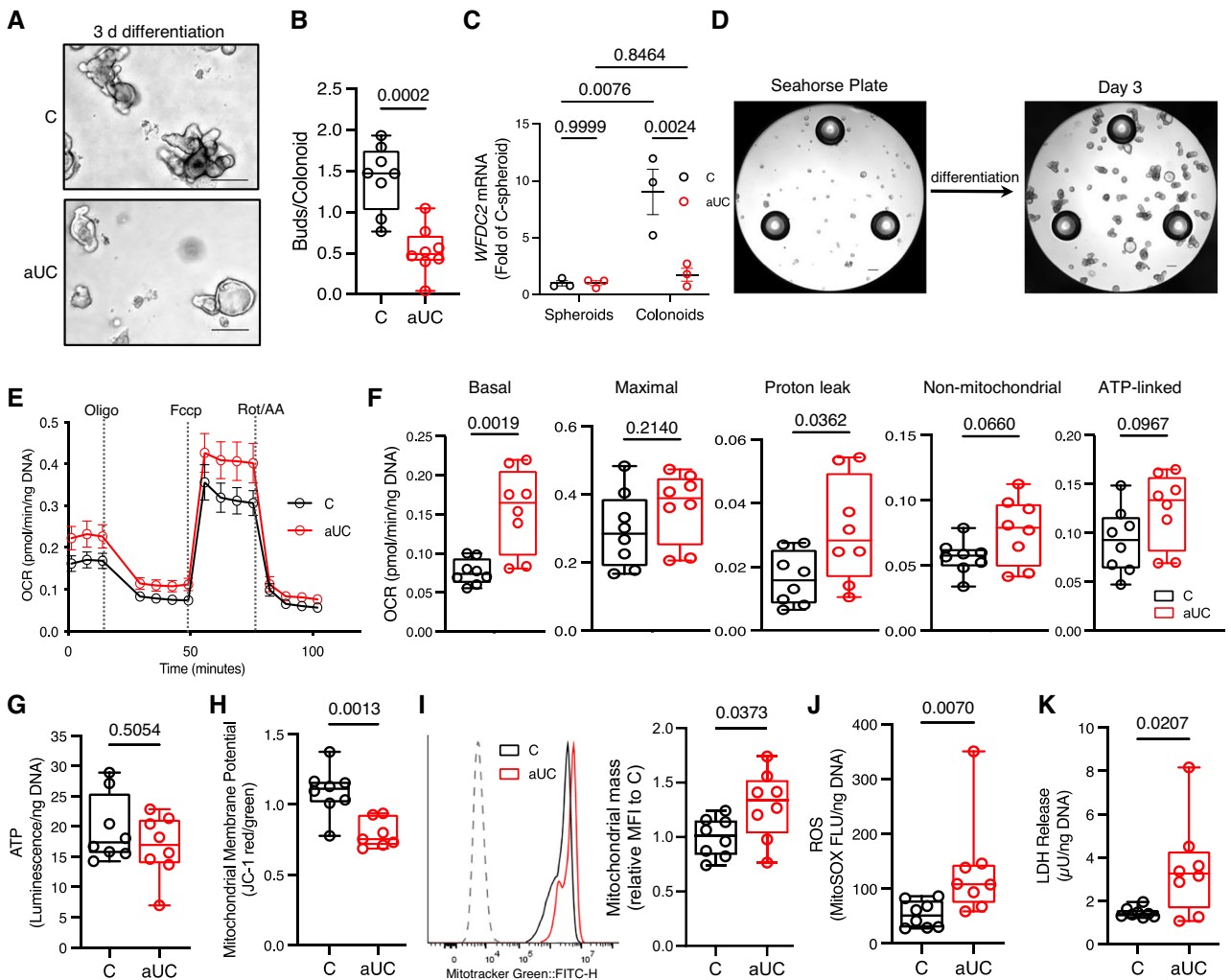

**Fig. 1 | Hypermetabolism and cellular stress in active UC colonoids during differentiation. A** Representative phase contrast images of pediatric control (C) and active (a)UC colonoids after 3-day differentiation. Scale bar, 200 μm. **B** Buds/colonoid in colonoids treated as in **A**. Each symbol represents one donor. $n = 8$ donors/group. **C** *WFDC2* mRNA abundance using qRT-PCR in paired spheroids (undifferentiated) and colonoids differentiated for 3 days. Each symbol represents one donor. $n = 3$ donors/gean ± SEM **D** Phase contrast image of a Seahorse 96-well plate with a representative spheroid line before and after 3-day differentiation. Scale bar, 200 μm. **E** Bioenergetic profile of C and aUC colonoids treated as in **A**, and subjected to the Seahorse MitoStress test. $n = 8$ donors/group, mean ± SEM **F** MitoStress OCR response of C and aUC colonoids. Each symbol represents an average of 3–4 replicates for one donor. $n = 8$ donors/group **G** Total ATP luminescence assay. Each symbol represents duplicate measures for one donor. $n = 8$ donors/group **H** Colonoids were treated as in **A**, and Mitochondrial membrane potential (MMP) estimation with the JC-1 dye by flow cytometry. Each symbol represents one donor. $n = 8$ donors (C) and 7 donors (aUC)/group **I** Representative histograms (left) and graph (right) of mitochondrial mass estimated with Mitotracker green fluorescence intensity in C and aUC colonoids treated as in **A**. Each symbol represents one donor. $n = 8$ donors/group **J** Mitochondrial ROS estimates in colonoids cultured as in **A**, with the MitoSOX fluorescence assay. Each symbol represents one donor. $n = 8$ donors/group **K** LDH activity in C and aUC colonoids after 2 days. Each symbol represents duplicate measures for one donor. $n = 8$ donors/group. Statistics: (**B**, **F**–**K**)· two-sided unpaired t-test with Welch correction in (**H**, **J** and **K**). **C**– 2-way ANOVA with Sidak *post hoc* test. For **B**, **F**–**K**, boxplot represents the first, second (median), and third quartiles with whiskers representing the minimum and maximum points. *P* values are indicated in the figures. LDH lactate dehydrogenase, OCR oxygen consumption rates, ROS reactive oxygen species.

obtained from inflamed epithelial regions, we wondered whether prior exposure to inflammation in vivo might explain the hypermetabolic nature of aUC colonoids in vitro. We exposed control and iUC spheroids to an inflammatory cocktail (TNFα, IL-1β, and flagellin) previously suggested to re-induce the inflammatory nature of UC in colonoids ex vivo[20] (Supplementary Fig. 4F). We observed that, unlike control colonoids, iUC spheroids showed reduced budded morphology after a 3-day differentiation when pre-exposed to the inflammatory cocktail. (Supplementary Fig. 4G). The inflammatory cocktail led to enhanced mtMass in iUC but not control colonoids (Supplementary Fig. 4H). Correspondingly, we found a significant increase in basal OCR in iUC colonoids pre-exposed to the inflammatory cocktail compared to vehicle and exposed control (Supplementary Fig. 4I and J). However,

there were no significant effects on maximal OCR and proton leak (Supplementary Fig. 4I and J). Collectively, these data suggest that iUC spheroids are more susceptible to inflammatory challenge and this leads to only a partial recapitulation of aUC-related enhanced respiratory phenotypes during differentiation.

## Metabolic uncoupling contributes to hypermetabolism in aUC colonoids

The proton leak and resulting metabolic inefficiency observed in aUC colonoids during differentiation may be indicative of enhanced uncoupling of mitochondrial respiration[24]. Thus, we hypothesized that endogenous mitochondrial uncoupling proteins would be upregulated in aUC compared to control. Analyses of bulk RNA-seq datasets

from the RISK study (GSE117993)[25] and PROTECT Pediatric IBD inception cohort studies (GSE109142)[14] showed that *UCP2* is significantly upregulated in the rectal mucosa of pediatric UC patients compared to controls (Supplementary Fig. 5A-C). Similarly, we observed enhanced UCP2 expression at the mRNA and protein levels in our patient-derived colonoid samples (Supplementary Fig. 5D and E).

To test whether metabolic uncoupling was sufficient to induce the altered colon epithelial differentiation and metabolism observed aUC, we added low-dose Fccp (protonophore that uncouples mitochondrial OXPHOS), into the medium throughout a 3-day differentiation of control colonoids (Supplementary Fig. 5F). Fccp raised basal OCR and OCR due to maximal respiration, proton leak, and non-mitochondrial respiration, but also significantly increased ATP-linked respiration, suggesting that the healthy colonoids may overcome chemical uncoupling to maintain ATP levels (Supplementary Fig. 5G and H). Still, we observed elevated mtMass and mitochondrial ROS levels in Fccp-treated colonoids compared to controls (Supplementary Fig. 5I and J). In contrast, we did not see an increase in LDH release to suggest uncoupling alone induced cytotoxicity (Supplementary Fig. 5K). Three days of Fccp treatment did not impact colonoid morphology or *WFDC2* gene expression (Supplementary Fig. 5L–N), but prolonged treatment up to day 6 resulted in a significant reduction in colonoid budding and WFDC2 expression (Supplementary Fig. 5P–R). These data suggest that uncoupling-induced hypermetabolic stress precedes impaired colonoid differentiation.

### Active UC colonoids exhibit an altered transcriptional profile and upregulated chemokine secretion

To understand transcriptional changes that may underly altered metabolism and function in aUC colonoids, we conducted bulk RNA-seq on spheroids and colonoids differentiated for 3 days ($n = 8$ patient lines per diagnosis, passage 5–7). Only seven genes were significantly upregulated and four were significantly downregulated in aUC compared to control undifferentiated spheroids (Log$_2$FC $\geq \pm 1$, FDR P $\leq 0.1$, Supplementary Data 1). After 3 days of differentiation, control and aUC colonoids had divergent transcriptomic profiles (Fig. 2A), in line with their distinct metabolic profiles. Specifically, 112 genes were significantly upregulated and 34 significantly downregulated (Log$_2$FC $\geq \pm 1$, FDR P $\leq 0.1$; Supplementary Data 1). The top upregulated genes included well-known MHCII-related genes in active UC[14,26], *HLA-DRA, HLA-DMB, HLA-DRB, HLA-DMA*; lipid-related genes (*FABP6, ABCA12, and APOL3*); and *CARD6* a member of the caspase recruitment domain family (Fig. 2B). Top downregulated genes included those involved in the development of cytoskeletal filaments (*NEFH, KRT12*), goblet cells (*ZG16, WFDC2*), and those involved in cellular development and histone maintenance (*GATA2* and *HIST1H4K*, respectively). Consistent with the metabolic profile of inactive UC colonoids and controls, only two genes (*HLA-DRA* and *HLA-DMB*) were significantly upregulated in iUC colonoids compared to control colonoids (Supplementary Data 1).

Chemokines recruit immune cells to the epithelium in UC, where they can incite damage. We observed enhanced gene expression of chemokine genes involved in the recruitment of macrophages (*CCL2*), T-cells (*CXCL11*), neutrophils, (*CXCL1*), and T- and B-cells (*CCL28*) in aUC colonoids compared to controls (Fig. 2C). Correspondingly, CCL2, CXCL11, and CCL1 proteins were disproportionately secreted into the medium in aUC compared to controls (Fig. 2D–F). We assessed the functional significance of the increased chemokine secretion by aUC colonoids by assessing the neutrophil chemoattractive potential in serum-free conditioned media from C and aUC colonoids (Fig. 2G). We observed that the conditioned medium from aUC colonoids had a more potent neutrophil chemoattractive potential than the medium from C colonoids (Fig. 2H), which suggests that increased chemokine secretion by aUC colonoids is functionally important for immune cell recruitment. We then determined whether metabolic uncoupling

during differentiation was sufficient to induce chemokine gene expression in control colonoids. Indeed, treatment of control colonoids with Fccp during differentiation significantly upregulated the gene of *CXCL1, CXCL11*, and *CCL2* on day 3 with persistent expression of *CXCL11, CCL2*, and *CCL28* on day 6 (Supplementary Fig. 5O and 4S). Thus, molecular events that stimulate metabolic uncoupling during epithelial differentiation in UC may contribute to the chemoattraction of immune cells.

### Active UC colonoids exhibit enhanced glycolytic capacity

To begin to understand the cellular underpinnings of hypermetabolism in aUC colonoids, we analyzed our differentially expressed transcriptomic data using functional annotation analysis (Ingenuity Pathway Analysis [IPA])[27] to predict the dominant molecular and cellular functional differences between aUC and control colonoids. IPA implicated carbohydrate metabolism as the top mediator of cellular function during differentiation (Fig. 3A, Supplementary Data 2). Measurement of real-time extracellular acidification rates (ECAR)[28] showed that aUC colonoids had an enhanced glycolytic capacity and reserve compared to control colonoids (Fig. 3B–E). Still, non-glycolytic acidification in aUC was significantly higher compared to controls, (Fig. 3F), which suggests other pathways such as $CO_2$ from the TCA cycle contribute to extracellular acidification[28]. To complement the extracellular flux analysis results, observed that on average, aUC colonoids consumed 4.2 times more glucose from the media and produced 2.8 times more lactate compared to controls over a 2-day differentiation (Fig. 3G, H). Collectively, the data indicate that, during the early stages of differentiation, aUC colonoids possess enhanced glycolytic capacity which could support the observed hypermetabolism.

### Dysregulated lipid metabolic genes in aUC colonoids

Functional annotation analysis of aUC colonoid differential gene expression also identified lipid metabolism as one of the top 10 dominant molecular and cellular functions (Fig. 3A). Upregulated lipid metabolism genes in aUC included those related to fatty acid transport (*FABP6, ABCA12*), apolipoproteins (*APOL1, APOL3*), sphingolipids and phospholipid biosynthesis (*SPTLC3* and *DGKG*, respectively), lipid-metabolizing CYPs (*CYPC218, CYP3A4*), and lipolysis (*LPL, PLB1, AADAC, LIPA*). We examined these genes in other colon bulk and single-cell RNA-seq data from UC patients. First, the PROTECT bulk RNA-seq data[14] showed enhanced expression of *ABCA12, APOL3, APOL1, LPL, DGKG*, and *LIPA* in active UC patients compared to controls (Supplementary Fig. 6A). Next, we analyzed publicly available UC single-cell RNA-seq data[29] to evaluate whether inflamed epithelial cells exhibit similar upregulation of lipid-related genes in UC. Compared to healthy samples, a higher proportion of the inflamed samples visibly expressed these identified lipid-related genes, including stem cells and other progenitor epithelial lineages (Supplementary Fig. 6B and C). Taken together, these analyses of the organoid, bulk tissue, and single-cell RNA-seq data sets indicate the transcriptomic dysregulation of lipid-related genes in UC which is modeled in our colonoids.

### Aberrant lipid accumulation and oxidation in aUC colonoids during differentiation

Considering the altered expression of lipid metabolic genes in aUC, we sought to determine whether enhanced lipid metabolism contributes to the observed hypermetabolic phenotype. Consistent with their analogous metabolic states before differentiation, control, and aUC spheroids had similar neutral lipid content (Fig. 4B). However, we observed a significant increase in neutral lipid accumulation during differentiation in aUC colonoids compared to their paired spheroids, while control spheroids and colonoids still retained similar neutral lipid content. (Fig. 4B). Furthermore, differentiated aUC colonoids contained elevated lipid content compared to the control colonoids (Fig. 4B, C). Our bulk RNA-seq data showed an upregulation of 4 lipid

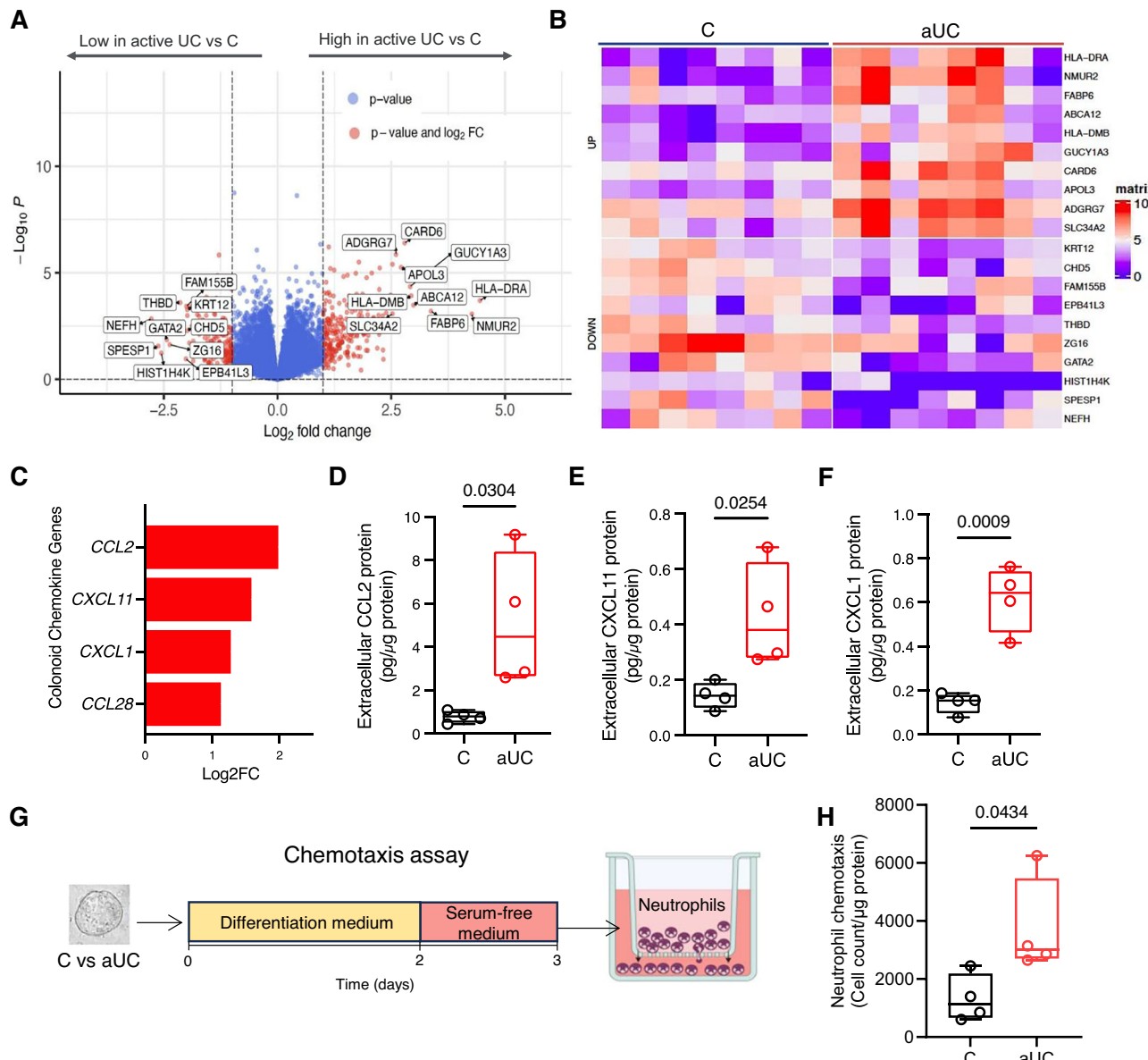

**Fig. 2 | Distinct transcriptome in epithelial colonoids from active UC pediatric patients.** Pediatric epithelial colonoids differentiated for 3 days and were subjected to bulk RNA-seq. $n = 8$ donors/group. **A** Volcano plot of differentially expressed genes between aUC and C samples (Log2FC > 1, FDR < 0.1). The full list of differentially expressed genes is provided in Supplementary Data 1. **B** Heat map of the top 10 dysregulated genes showing the diversity and consistency of expression among each colonoid line. **C** Chemokine genes overexpressed in aUC epithelial colonoids compared to C in the bulk RNA-seq data (Log2FC > 1, FDR < 0.1). **D** Extracellular chemokine protein, CCL2, **E** CXCL11, and **F** CXCL1 after a 3-day differentiation. Each symbol represents duplicate measures for one donor. $n = 4$ donors/group. **G** Schematic of chemotaxis assay. Partly created in BioRender. Ojo, B. (2025) https://BioRender.com/nwwqemf **H** Neutrophil chemotaxis after 2 hours with conditioned medium. Each symbol represent the average duplicate measures for one donor. $n = 4$ donors/group. Statistics for **D**–**F** and **H**: two-sided unpaired t-test. For **D**–**F**, and **H**, boxplot represents the first, second (median), and third quartiles with whiskers representing the minimum and maximum points. *P* values are indicated in the figures.

hydrolysis genes (*LPL, PLB1, AADAC, LIPA*) in aUC colonoids (Fig. 4A) suggesting that hydrolysis of accumulated lipids to free fatty acids may undergo fatty acid oxidation (FAO) and support OXPHOS. We tested the protein activity of one of the lipid hydrolysis genes– LPL, and found elevated LPL activity in aUC colonoids compared to control on Day 2 with no significant difference between groups by Day 5 (Fig. 4D). These data further suggest that active lipid cycling may be overamplified during differentiation in the UC colon epithelium and our pediatric colonoid model of UC may recapitulate these lipid metabolic dynamics.

To characterize differential lipid abundance in aUC colonoids in more detail, we performed large scale targeted lipidomics assessing 762 lipid species. We observed 96 differentially abundant lipid species from 15 classes between differentiated aUC and C colonoids (FC ≥ 2, FDR ≤ 0.05). Interestingly, all 96 statistically significant lipids were more abundant in aUC compared to C (Fig. 4E). Major overabundant lipid classes in aUC include 20 mitochondrial-bound acylcarnitines, 13 ceramides, 5 tri- and 4 di-hexosylceramides, 29 phosphatidylethanolamines and 8 phosphatidylinositols. The 50 most differentially abundant lipids are shown in Fig. 4F and the full list in Supplementary Data 3.

Subsequently, we hypothesized that the oversupply of lipids contributes to elevated $O_2$ consumption in aUC colonoids during differentiation. Thus, we exposed control and aUC colonoids to a

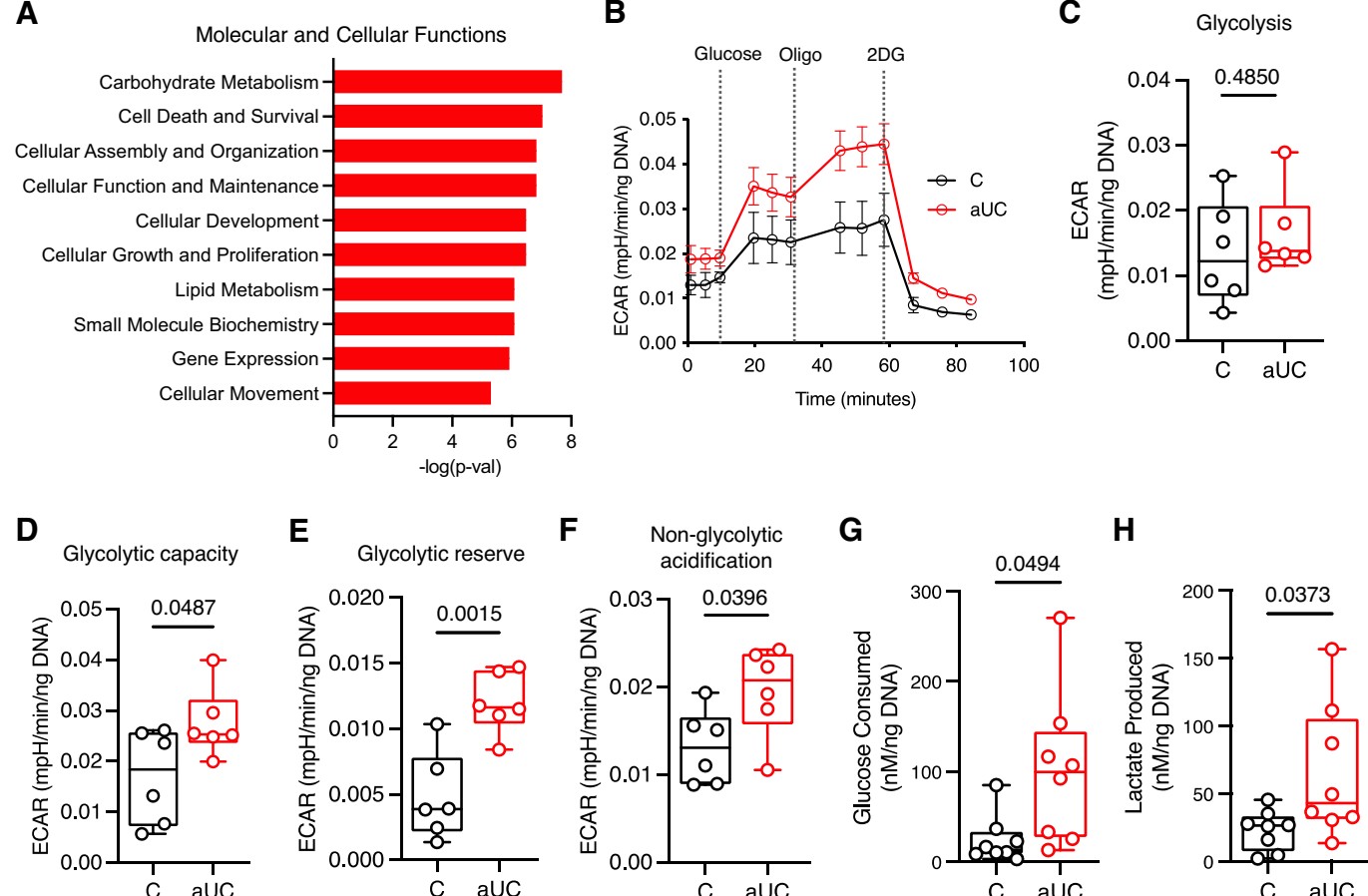

**Fig. 3 | Higher glycolytic capacity in active UC colonoids during differentiation.** **A** Ingenuity Pathway Analysis of differentially expressed RNAs between aUC and C colonoids. The top 10 Molecular and Cellular Functions are shown. Full details of molecules involved under each function are presented in Supplementary Data 2. **B** Bioenergetic profile of C and aUC colonoids differentiated in 96-well Seahorse plates for 3 days and subjected to the Seahorse Glycostress assay. $n = 6$ donors/group, mean ± SEM. **C**–**F** Glycostress extracellular acidification rates (ECAR) of C and aUC colonoids as in (**B**) to determine real-time (**C**) glycolysis, (**D**) glycolytic capacity, (**E**) glycolytic reserve, and (**F**) non-glycolytic acidification rate. Each symbol represents an average of 4 replicates for one donor. $n = 6$ donors/group. **G** Extracellular glucose and **H** extracellular lactate concentration in the medium from C and aUC colonoids differentiated for 2 days. Each symbol represents an average of duplicate readings for one donor. $n = 8$ donors/group. Statistics: (**A**) Right tailed Fisher's Exact Test (**C**–**H**)- two-sided unpaired t-test, with Welch correction in **G** and **H**. For **C**–**H**, boxplot represents the first, second (median), and third quartiles with whiskers representing the minimum and maximum points. *P* values are indicated in the figures. ECAR, extracellular acidification rates.

substrate-depleted medium with or without etomoxir, a carnitine palmitoyl-transferase 1a (CPT1a) inhibitor that inhibits fatty acid oxidation (Fig. 4G). With negligible external sources of metabolic substrates, aUC colonoid mean basal OCR was 78% greater than that of controls (Fig. 4H, I). Etomoxir reduced basal OCR in aUC colonoids to the level of control colonoids (Fig. 4I). Etomoxir treatment significantly reduced proton leak in aUC (Fig. 4J). We also observed etomoxir to reduce proton leak in C colonoids, but the difference was not statistically significant. Compared to C colonoids, we observed a significant increase in ATP-linked OCR in aUC colonoids under this nutrient-depleted condition, with a significant decrease in ATP-linked OCR in aUC with etomoxir inhibition (Fig. 4K). Etomoxir did not impact maximal respiration and non-mitochondrial respiration (Supplementary Fig. 7A and B). These results indicate that enhanced lipid accumulation and oxidation in the active UC epithelium is necessary for the hypermetabolic phenotype during differentiation.

### Lipotoxicity precedes *CXCL1* overexpression during colonoid differentiation

Lipid exposure induces stemness in intestinal progenitor cells[30]; hence, we wondered whether lipotoxicity during colonoid differentiation may

contribute to the impaired differentiation and chemokine expression observed in aUC colonoids (Figs. 1C, 2C). Thus, we exposed control spheroids to BSA-bound palmitic acid (BSA-PA) during the initial 24 hrs of differentiation. We found a dose-dependent increase in the gene expression of the uncoupling protein *UCP2* and the fatty acid transporter *FABP6*, (Supplementary Fig. 7C) in response to BSA-PA exposure. In four control donor lines, we confirmed a significant modest upregulation of *UCP2* and *FABP6* with the highest tested PA dose (Supplementary Fig. 7D). Consequently, we hypothesized that lipid-induced stress to the colon epithelial progenitors in aUC might contribute to the enhanced chemokine production we observed in aUC colonoids under non-stimulated conditions in vitro (Fig. 2C–F). We pre-exposed control colonoids to PA for 3 days with further differentiation without PA till day 5. Compared to BSA-EtOH control, PA exposure impaired colonoid morphology (Supplementary Fig. 7E and F) and induced a significant upregulation of *CXCL1* in all colonoid lines (Fig. 4L), with donor-dependent responses for other chemokine genes. Interestingly, blocking mitochondrial lipid oxidation with etomoxir during the 3-day PA exposure reduced CXCL1 protein secretion (Fig. 4M, N), suggesting that mitochondrial β-oxidation is important for such inflammatory response. Overall, these data suggest that fatty acid overexposure at the early stages of differentiation

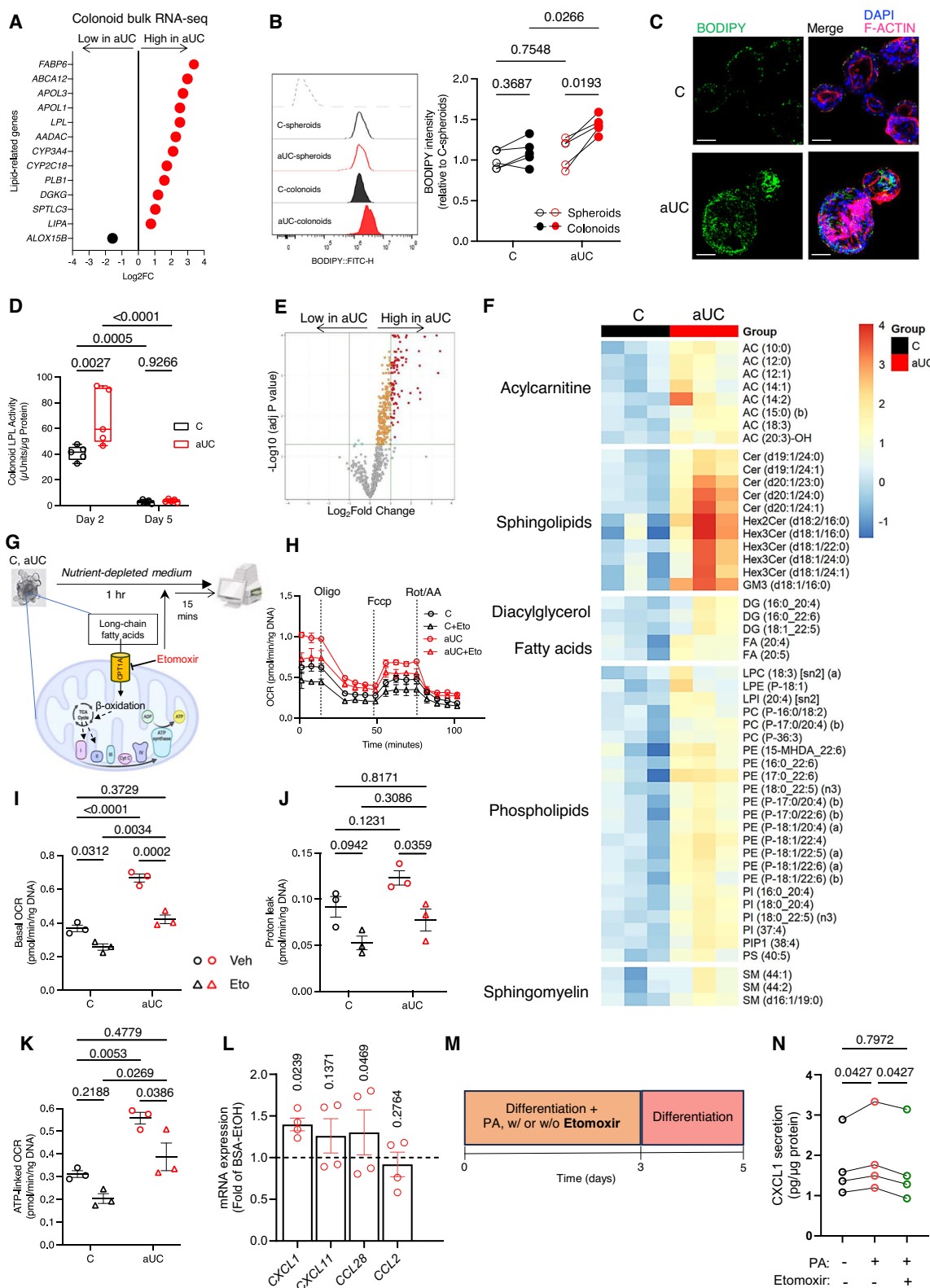

may contribute to inflammation and impair differentiation in epithelial colonoids.

## PPAR-alpha contributes to hypermetabolic stress during colonoid differentiation

To understand the regulatory networks that may explain the hypermetabolic nature of differentiating aUC colonoids, we conducted a Causal Network Analysis[27] of our colonoid bulk transcriptomic data. Among other notable molecular networks, the PPARA-RXRA network was the most significant causal factor (Fig. 5A, Supplementary Data 4). Analyses of lineage-specific *PPARA* expression in the scRNA-seq data from treatment-naïve UC colon epithelium[29] showed the upregulation of *PPARA* majorly in progenitor cells (stem, TA2, cycling TA, secretory TA) in the inflamed and non-inflamed colon epithelial regions

**Fig. 4 | Dysregulated lipid metabolism supports hypermetabolism in active UC colonoids during differentiation. A** Differentially expressed lipid-related genes (FDR < 0.1) in aUC vs C epithelial colonoids differentiated for 3 days. Dots represent the Log2 fold change in aUC compared to C colonoids. $n = 8$ donors/group. **B** BODIPY+ neutral lipid estimation in undifferentiated (spheroids) and 2-day differentiated colonoids by flow cytometry. $n = 5$ donors/group **C** Representative confocal images of C and aUC colonoids of BODIPY+ neutral lipid accumulation. Scale bar, 50 μm. **D** LPL protein activity in C and aUC colonoids after a 2-day and 5-day differentiation. $n = 5$ donors/group. Boxplot represents the first, second (median), and third quartiles with whiskers representing the minimum and maximum points. **E** Volcano plot from lipidomics analysis of aUC vs C colonoids differentiated for 3 days showing 96 lipid metabolites differentially upregulated (red dots) in aUC vs C. $n = 3$ donors/group **F** Heatmap of the top 50 lipid classes differentially abundant in aUC vs C colonoid lipidomics analysis. $n = 3$ donors/group. **G** Schematic of Etomoxir treatment in C and aUC colonoids differentiated for

3 days. Partly Created in BioRender. Ojo, B. (2025) https://BioRender.com/nwwqemf . **H** Bioenergetic profile of C and aUC colonoids as in (**G**). $n = 8$ donors/group, mean ± SEM **I** Basal OCR, **J** Proton leak, and **K** ATP-linked OCR response of C and aUC colonoids as in **G**. $n = 8$ donors/group. **L** Chemokine genes in non-IBD colonoids exposed to 200 μM BSA-PA or BSA-EtOH for 3 days before further differentiation up to 5 days. $n = 4$ donors/group, mean ± SEM. **M** Schematic of colonoids from non-IBD donors exposed to BSA-EtOH control or 200 μM BSA-PA with or without etomoxir for 3 days before further differentiation up to 5 days. **N** Extracellular CXCL1 in the last 2 days of non-IBD colonoids treated as in **M**. $n = 4$ donors/group. Statistics: (**B**, **D**, **I**–**K**)- Two-way ANOVA with Tukey's test in **B**, **I**–**K**, and Holm-Sidak test in **D**. **E**- Moderated T-test with Benjamini-Hochberg correction (**L**)- two-sided paired t-test. **N**- RM One-way ANOVA with Holm-Sidak test. $P$ values are indicated in the figures. BSA bovine serum albumin, Eto etomoxir, OCR oxygen consumption rates, PA, Palmitic acid, Veh vehicle.

compared to healthy samples (Supplementary Fig. 8A). We did not see any differences in *PPARA* mRNA expression in our bulk RNA-seq data between aUC and control colonoids differentiated for 3 days; however, we found an enhanced PPAR-α activity in nuclear extracts from aUC colonoids compared to controls at the early stages (24 hr) of differentiation (Supplementary Fig. 8B).

We wondered whether PPAR-α activation is sufficient to induce the observed hypermetabolic phenotype. Treatment of control colonoids with the PPAR-α agonist– fenofibrate during differentiation significantly reduced colonoid budded morphology and downregulated the crypt-base goblet cell marker[23] *WFDC2* (Fig. 5B-E). Fenofibrate enhanced glucose consumption vs vehicle which was not accompanied by significant lactate production, suggesting a negligible induction of anaerobic glycolysis (Fig. 5F, G). Fenofibrate treatment enhanced UCP2 expression during colonoid differentiation (Fig. 5H). Consistent with aUC lipid metabolic phenotypes, neutral lipid droplets, and LPL activity increased in control colonoids exposed to fenofibrate (Fig. 5I–K). Fenofibrate enhanced mitochondrial mass, MitoSOX ROS, with a resultant elevation in basal, maximal, proton leak, and non-mitochondrial OCR with no significant increase in ATP-linked OCR (Fig. 5L–O). Finally, fenofibrate treatment induced LDH release consistent with our observation in differentiating aUC colonoids (Fig. 1K, Fig. 5P). These findings suggest that increased PPAR-α activity during colon epithelial differentiation significantly contributes to hypermetabolic stress.

### Pharmacological inhibition of PPAR-α reprograms epithelial metabolism in aUC colonoids

Following the observed contribution of PPAR-α activity to lipid accumulation and hypermetabolic stress during differentiation, we reasoned that inhibition of PPAR-α during aUC colonoid differentiation may ameliorate hypermetabolic stress. We treated aUC colonoids with the PPAR-α antagonist, GW6471 (GW), during differentiation. GW treatment suppressed neutral lipid accumulation and downregulated *LPL* and *UCP2* genes compared to untreated aUC colonoids (Fig. 6A–C, Supplementary Fig. 9A). Consequently, in a substrate-enriched medium (10 mM glucose, 1 mM pyruvate, 2 mM glutamine), we observed a significant reduction in basal, maximal, proton leak, and ATP-linked OCR in GW-treated colonoids (Fig. 6D, E). Luminescence assay for total cellular ATP showed no significant differences in untreated and GW-treated aUC colonoids (Fig. 6F). These data suggest that PPAR-α inhibition reduced lipid accumulation and improved cellular bioenergetics in active UC colon epithelial cells. Under substrate-depleted conditions, blockade of FAO with etomoxir did not affect cellular bioenergetics measures in GW-treated aUC colonoids (Fig. 6G, H, Supplementary Fig. 9B–E) which confirmed that GW abrogates the contribution of lipid accumulation and oxidation to cellular bioenergetics in aUC colonoids. Consequently, we observed a significant increase in glucose consumption in all GW-treated aUC colonoid lines

compared to untreated aUC colonoids without changes in lactate production, suggesting an increased reliance on glucose as a substrate to drive cellular metabolism (Fig. 6I, J).

We next examined whether the metabolic reprogramming in aUC colonoids with PPAR-α inhibition impacted cellular health markers. GW treatment in aUC colonoids did not impact mitochondrial mass or mitochondrial ROS levels compared to untreated aUC colonoids (Supplementary Fig. 9F and G). In contrast, GW treatment significantly reduced LDH release (cytotoxicity marker) in every colonoid line compared to untreated aUC colonoids (Fig. 6K). In the medium collected from aUC colonoids differentiated for 2 or 5 days, we found that GW treatment significantly suppressed extracellular CXCL1 (Fig. 6L). There was no significant effect of GW treatment on extracellular CXCL11 and a trend toward a reduction in extracellular CCL2 on day 2 ($p = 0.053$), while there was no effect on day 5 (Supplementary Fig. 9H and I). Blinded visual assessment of colonoid morphology showed no consistent impact of GW on colonoid budding (Supplementary Fig. 9J and K). The gene expression of stemness markers (*LGR5* and *ASCL2*) reduced over time in every aUC colonoid line with GW treatment, with significant downregulation in *ASCL2* (Fig. 6M). *WFDC2* was upregulated in GW-treated colonoid lines compared to untreated aUC colonoids while the increase in *MUC2* expression was not statistically significant (Fig. 6M). Overall, these data suggest that PPAR-α inhibition in aUC epithelial colonoids induced a metabolic shift away from lipid oxidation, which consequently reduced chemokine secretion and augmented molecular markers of epithelial differentiation.

## Discussion

Leveraging clinical samples to develop human IBD pre-clinical models is important for determining mechanisms of human disease pathogenesis. We used patient-derived colon epithelial organoids as a human model to study epithelial metabolic dysfunction in pediatric UC, since these organoids may recapitulate the molecular features of the parent tissue. We report profound elevation in oxygen consumption in aUC colonoids during differentiation related primarily to increased proton leak. This hypermetabolic phenotype was associated with increased chemokine secretion, oxidative stress, cytotoxicity, and impaired differentiation. The aUC colonoids showed an elevated capacity for lipid accumulation during differentiation and their hypermetabolic phenotype required FAO. Our data indicate the contribution of the PPAR-α pathway in driving altered lipid metabolism and OXPHOS in aUC colonoids and that pharmacological inhibition of PPAR-α alleviates abnormal metabolism, chemokine overexpression, and differentiation in aUC epithelial cells.

In our study, colonoids from active UC patients showed increased mitochondrial respiration during differentiation without a concomitant change in ATP levels. The normal colon epithelium is less dependent on highly oxidative processes for energy generation[31,32].

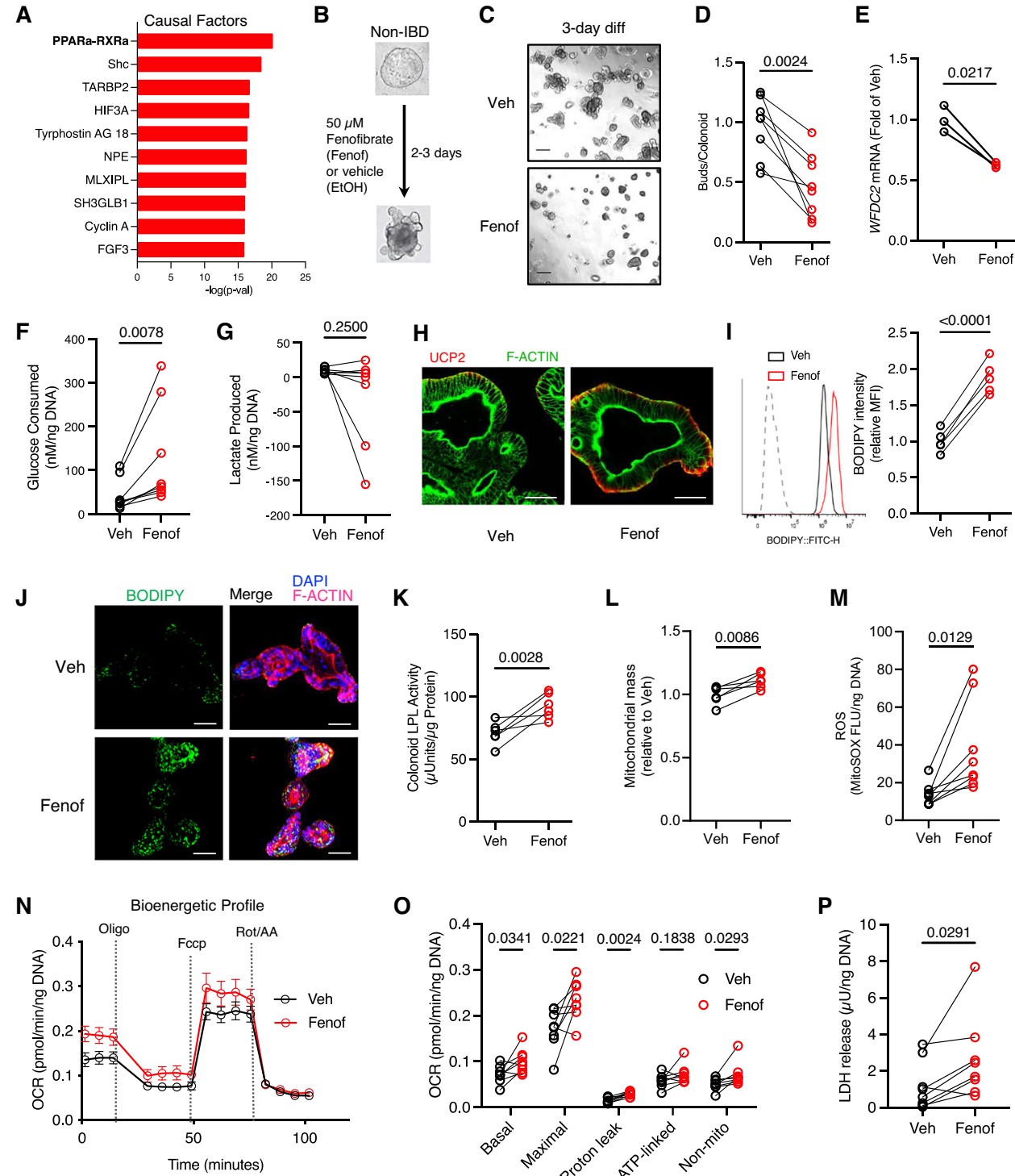

The hypermetabolic and inflammatory phenotype we observed in aUC colonoids is largely dependent on lipid overaccumulation and fatty acid oxidation. We found that overexposure of control colonoids to long-chain fatty acids alone increased expression of the uncoupling protein *UCP2* and the chemokine *CXCL1*, and that oxidative uncoupling alone was sufficient to induce chemokine expression. Finally, inhibition of FAO reversed hypermetabolism and reduced chemokine production in aUC colonoids. Thus, the overaccumulation of several lipid species in aUC colonoid may tie mitochondrial uncoupling and hyperactivity to higher chemoattractive potential observed in aUC colonoids. Indeed, hypermetabolism may channel energetic demands towards molecular programs that favor a hypersecretory phenotype[33,34], which may explain the hypersecretion of chemokines in aUC colonoids during differentiation. Furthermore, the oversupply of metabolic substrates may result in hypermetabolism with associated mitochondrial and cellular stress[35]. Cellular overdependency on lipids for energy generation is inefficient since the oxidation of lipids consumes energy and induces metabolic uncoupling[36] as observed with increased proton leak in our aUC colonoids. Aberrant lipid exposure or lipid droplet hydrolysis may trigger lipotoxicity-related pathways including apoptosis, inflammation, and ER stress[37]. Overall, our data suggest a lipid-related hypermetabolism during differentiation in epithelial aUC colonoids, which may increase the energetic cost of differentiation and cellular stress in aUC. Hence, it is plausible that

**Fig. 5 | PPAR-α agonism in non-IBD colonoids induces hypermetabolic stress during differentiation. A** Ingenuity Pathway Analysis of bulk transcriptomic data showing the top 10 predicted Causal Networks of differential gene expression between aUC and C colonoids. **B** Schematic of PPARA agonist, fenofibrate, treatment in non-IBD colonoids during differentiation. **C** Representative phase contrast images of pediatric C colonoids treated as in **B**. Scale bar, 200 μm. **D** The number of buds/colonoid in colonoids treated as in **B**, and differentiated for 3 days. Each symbol represents one donor. $n = 8$ donors/group. **E** *WFDC2* gene expression in colonoids treated as in **B**, assessed by qRT-PCR. Each symbol represents duplicate measures of one donor. $n = 3$ donors/group. **F** Extracellular glucose and **G** extracellular lactate concentration in medium from C colonoids treated with or without fenofibrate. Each symbol represents duplicate measures of one donor. $n = 8$ donors/group. **H** Representative confocal immunofluorescence images of UCP2 expression in non-IBD colonoids treated as in **B**, for 3 days. Scale bar, 50 μm **I** BODIPY+ neutral lipid estimation by flow cytometry. Each symbol represents one

donor. $n = 5$ donors/group. **J** Representative confocal images of C colonoids visualizing BODIPY+ neutral lipids. Scale bar, 50 μm. **K** LPL activity in C colonoids treated as in **B**, and differentiated for 2 days. Each symbol represents duplicate measures of one donor. $n = 6$ donors/group. **L** Mitochondrial mass in C colonoids treated as in **B**, and differentiated for 3 days. Each symbol represents one donor. $n = 6$ donors/group. **M** MitoSOX ROS estimate in C colonoids treated as in **B**, for 3 days. Each symbol represents duplicate measures of one donor. $n = 8$ donors/ group. **N** Bioenergetic profile of C colonoids treated as in **B**, after a 3-day differentiation. $n = 8$ donors/group, mean ± SEM. **O** MitoStress OCR responses of C colonoids in **N**. Each symbol represents an average of 3–4 replicates of one donor. $n = 8$ donors/group. **P** LDH activity in the medium during differentiation. Each symbol represents duplicate measures of one donor. $n = 8$ donors/group. Statistics: (**D, E, I, K–M, O, P**)- two-sided paired t-test; (**F, G**)- two-sided Wilcoxon signed-rant test. *P* values are indicated in the figures. LDH, lactate dehydrogenase; OCR, oxygen consumption rates; Veh, vehicle.

---

approaches that suppress aberrant epithelial lipid accumulation, hydrolysis, and oxidation may improve colon epithelial function in UC.

The large PROTECT pediatric UC inception cohort study[14] found extensive downregulation of genes related to mitochondria and OXPHOS in the rectal mucosa. While we did not observe any significant changes in mitochondrial genes in our aUC colonoids, we observed abnormally high mitochondrial respiration in aUC colonoids. Similarly, enhanced epithelial OXPHOS activity was reported in a mouse model of colitis during the active stage of the disease at day 7 despite the downregulation of some mitochondrial genes[38]. These metabolic and transcriptional disparities in the active epithelium largely resolved to the level of control upon recovery at day 28[38]. There may be other contributors to epithelial metabolic dysregulation in UC, including the colon mucosa's inflammatory milieu, and it may be that our colon organoids predominantly retained the lipid-dependent components in culture.

Our study also suggests a role for the PPAR-α network in the UC epithelium and colonoids. PPAR-α activation in non-IBD colonoids during differentiation recapitulated lipid-induced hypermetabolic stress, morphologic and molecular evidence of impaired differentiation, similar to our observation in aUC colonoids. Consistent with our study, PPAR-α agonism exaggerated colitis in DSS-induced mouse models with a concomitant increase in bioactive serum lipids[39,40]. Furthermore, activation of fatty acid oxidation partly via PPAR-α promotes intestinal stemness in high-fat diet models[30], suggesting a detrimental effect on intestinal stem cell differentiation and maintenance of epithelial homeostasis. Natural PPAR-α agonists including PUFAs are disproportionally accumulated in the inflamed colon mucosa of UC patients which correlated positively with endoscopic disease activity[41,42]. Sphingolipids may induce lipid oxidation via PPAR-α[43] and bacterial-derived sphingolipids enhance susceptibility to DSS-induced colitis[44]. Our colonoid lipidomics analysis revealed an over-accumulation of 2 PUFAs, 7 acylated PUFAs and 26 sphingolipid species in aUC. Considering that lipids are known to induce metabolic uncoupling in other tissues[45–47] similar to the outcomes in our study, we hypothesize that the overexposure of colon epithelial stem cells to dietary or bacterial-derived LCFAs, PUFAs, or complex lipids in vivo will promote epithelial hypermetabolic phenotypes, uncoupling, and epithelial metabolic inefficiency during colon epithelial differentiation. Whether this happens in a PPAR-α-dependent manner in vivo could be the subject of future studies.

In contrast, pharmacological inhibition of PPAR-α in our aUC colonoids suppressed aberrant neutral lipid accumulation and oxidation. In this scenario, aUC colonoids consumed more glucose, likely due to a metabolic switch to compensate for the loss of lipid-dependent oxidation for energy, resulting in reduced cytotoxicity and CXCL1 secretion. Hypermetabolism may divert energy expenditure towards translational processes that upregulate cytokine and stress

responses[33]; hence, it is plausible that the blockade of lipid's contribution to the hypermetabolic state in aUC colonoids by PPAR-α inhibition reduces such cytotoxic stress responses. Animal studies have suggested the potential of PPAR-α inhibition in ameliorating colitis[48,49]. In our study, PPAR-α induction in control colonoids impaired transcriptomic and morphologic measures of differentiation and PPAR-α inhibition improved transcriptomic measures of differentiation in aUC colonoids, indicating PPAR-α activity inhibits epithelial differentiation. However, we did not observe improvement in colonoid budded morphology with PPAR-α inhibition, indicating the potential involvement of other pathways in colonoid morphological development in aUC. Overall, our study in human patient-derived colonoids suggests that PPAR-α antagonism in the epithelium may be a viable strategy for reducing the contribution of lipotoxicity to hypermetabolic epithelial stress in UC.

In conclusion, our data in patient-derived epithelial colon organoids indicate that hypermetabolism and lipid metabolic dysregulation significantly contribute to colon epithelial dysregulation in pediatric UC. Furthermore, epithelial PPAR-α may be a therapeutic target for restoring colon epithelial homeostasis by reducing dysregulated lipid metabolism and normalizing cellular respiration. Overall, the use of patient-derived colonoids in this study opens future research directions in colon epithelial metabolism that may identify metabolic targets for next-generation therapies specifically directed toward healing the colon epithelia in pediatric UC.

## Methods
### Study design and clinical samples
Pediatric patient recruitment occurred at Cincinnati Children's Hospital Medical Center, Cincinnati, Ohio, and Lucile Packard Children's Hospital, Stanford, Palo Alto, California. Protocols for recruitment and biopsy collection were approved by the Institutional Review Board (IRB) at Cincinnati Children's Hospital Medical Center and Stanford University. Consent was obtained from parents or legal guardians, and age-appropriate assent was obtained from each patient donor. Sex assigned at birth was recorded for each participant, but did not influence enrollment. Four to eight colon biopsies (2–5 mm³) were obtained during lower endoscopy at inflamed rectal sites of patients with active UC, with location-matched biopsies obtained from inactive UC, and non-IBD controls. Biopsies were collected into ice-cold DMEM/ F12 (Corning 10-092-CV) + 50 μg/mL Normocin (InvivoGen #ant-nr) and transported on ice to the lab for immediate processing for organoid generation. Donor demographics and clinical data are presented in Supplementary Table 1.

### Biopsy processing and organoid culture
Fresh biopsies were transferred into 5 mL of 2 mM PBS-EDTA and incubated for 45 min on a rotator in a cold room. Biopsies were

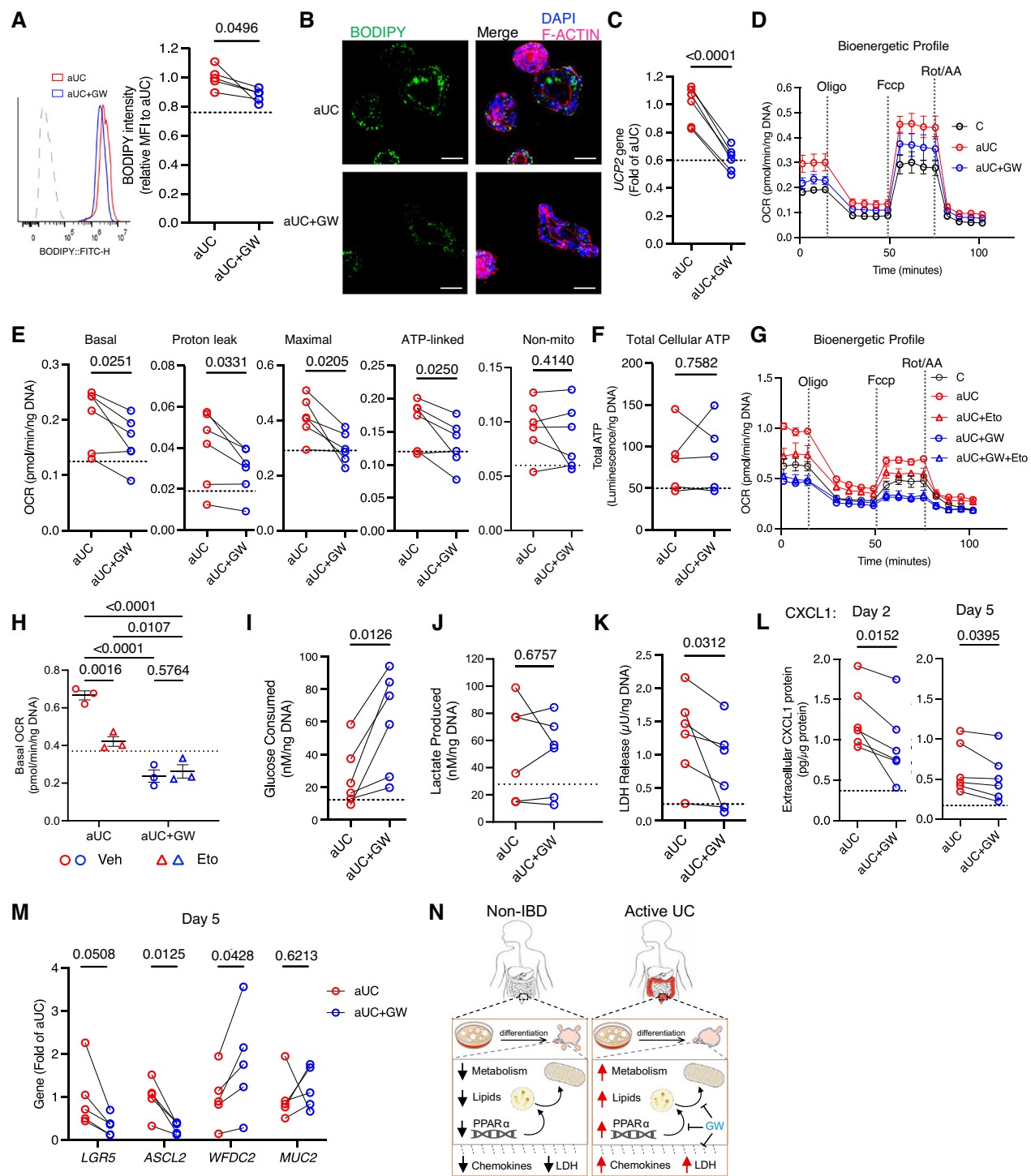

transferred into 2.5 mL ice-cold sorbitol buffer (2% sorbitol, 1% sucrose, 1% BSA, 50 μg/mL Normocin) in a petri dish and crypts were gently scraped off using a college cotton tweezer under a dissecting microscope (Leica S9E). Big tissue pieces were removed with a 20 μL pipette tip and the crypts and buffer were transferred into a 15 mL tube. Crypts were sheared briskly with a 1000 μL pipette (80X total with 1 min interval) and the suspension was made up to 14 mL with ice-cold PBS (Gibco #10010-023). The suspension was centrifuged (300 x *g*, 5 mins, 4 °C) and the pellet was resuspended in organoid wash medium, OWM, (DMEM/F12, 10% FBS, 50 μg/mL Normocin, and washed twice (300 x *g*, 5 mins, 4 °C). The visible pellet was resuspended in growth factor-reduced Matrigel (Corning #356231) and 50 μL drops of the Matrigel

mixture were allowed to polymerize at 37 °C for at least 15 mins on pre-warmed 12-well plates. Upon stabilization, 700 μL of Intesticult organoid growth medium, OGM, (STEMCELL Technologies, #06010) containing 10 μM TGF-beta receptor inhibitor, SB431542 (Cayman Chemical, #13031), 10 μM ROCK inhibitor, Y27632 (STEMCELL Technologies, #72304), and 50 μg/mL Normocin. Crypts were placed in a tissue culture incubator (37 °C, 5% $CO_2$) and OGM was replaced every 2–3 days for the first 7–14 days. Spheroids were considered stable and mature enough for expansion when >70% of spheroids are ~100 μm in diameter.

For spheroid expansion, OGM was aspirated and 1 mL of ice-cold PBS was added to each well. The Matrigel domes were

**Fig. 6 | PPAR-α inhibition reprograms epithelial metabolism and ameliorates hypermetabolic stress in active UC colonoids during differentiation. A** Flow cytometry of BODIPY⁺ neutral lipids in aUC colonoids treated with 1 μM of the PPAR-α antagonist, GW6471 (GW), or vehicle (EtOH) during differentiation. Each symbol represents one donor. *n* = 5 donors/group. **B** Representative confocal images of aUC colonoids treated as in **A**, for visualizing BODIPY⁺ neutral lipid accumulation. Scale bar, 50 μm. **C** UCP2 gene expression by qRT-PCR. Each symbol represents one donor. *n* = 6 donors/group. **D** Bioenergetic profile of aUC colonoids treated with or without GW and subjected to the Seahorse MitoStress test. *n* = 6 donors/group, mean ± SEM. **E** MitoStress OCR responses of aUC colonoids treated as in **D**. Each symbol represents an average of 3–4 replicates of one donor. *n* = 6 donors/group. **F** ATP luminescence assay in colonoids cultured as in **D**. Each symbol represents an average of duplicate measures of one donor. *n* = 5 donors/group. **G** Bioenergetic profile of aUC colonoids subjected to the Seahorse MitoStress test in a nutrient-deprived medium with or without etomoxir. *n* = 3 donors/group, mean ± SEM.

**H** Basal OCR of aUC colonoids as in **G**. Each symbol represents an average of four replicates of one donor. *n* = 3 donors/group, mean ± SEM. **I** Extracellular glucose and **J** lactate in the medium after 2 days. Each symbol is an average of duplicate measures of one donor. *n* = 6 donors/group. **K** LDH activity after a 2-day differentiation sampled from the same plate as **I**. Each symbol is an average of duplicate measures of one donor. *n* = 6 donors/group. **L** Differentiation medium collected on day 2 and on day 5 was used to assess CXCL1 secretion. Each symbol is an average of duplicate measures of one donor. *n* = 6 donors/group. **M** Effect of GW on gene expression of differentiation markers assessed by qRT-PCR. Each symbol represents one donor. *n* = 5 donors/group. **N** Proposed model of epithelial metabolic dysregulation in pediatric UC colonoids. Icons are from NIAID NIH BioArt. Statistics: (**A, C, E, F, I, J, L, M**)- two-sided paired t-test; **H**- Two-way ANOVA, Tukey post hoc test; **K**- Wilcoxon signed-rank test. *P* values are indicated in the figures. Eto, etomoxir OCR, oxygen consumption rates; Veh, vehicle.

---

dislodged from the bottom of the plate with a 1000 μL pipette and the PBS-Matrigel mixture was transferred into a 15 mL tube. The Matrigel was broken up inside the tube with a 1000 μL pipette, made up to 5 mL with ice-cold PBS, and incubated on ice for 5 min. The suspension was centrifuged (300 x g, 5 mins, 4 °C), 350 uL of TrypLE Express (Gibco #12604-013) was added to the spheroid pellets and incubated in a 37 °C water bath for 4-5 mins. Spheroids were broken up by pipetting 50X with a 1000 μL pipette followed by the addition of 5 mL ice-cold OWM. Spheroids were washed twice in OWM (300 x g, 5 mins, 4 °C), the resulting pellets were resuspended in 3 parts of Matrigel to 1 part of OWM, and 50 μL drops of the Matrigel suspension were plated in pre-warmed 12-well plates and allowed to polymerize. For spheroid expansion, 2–3 drops of this suspension were plated per well of a 12-well plate and 800 μL of 50% L-WRN conditioned medium[50,51] containing 10 μM TGF-beta receptor inhibitor, SB431542 and 10 μM ROCK inhibitor, Y27632 and replaced every other day.

For cryopreservation, spheroids underwent at least 3 passages before storage in our pediatric organoid biobank. Matrigel domes with matured spheroids (~80 μm) were dislodged in ice-cold PBS and centrifuged (300 x g, 5 mins, 4 °C). Pellets were resuspended in freezing medium (90% FBS, 10% DMSO), and 1 mL of the suspension was added to a cryotube. Samples were slow-frozen in a Mr. Frosty freezing container at −80 °C and transferred to liquid N₂ for long-term storage. Overall, the pooling of matured spheroids from two Matrigel domes in one cryotube allows for sufficient post-thaw recovery of spheroids.

All experiments used cryopreserved spheroids. Withdrawal of cryotubes from the LN₂ tank followed a quick thaw in a 37 °C water bath for ~2 mins. Cryotube contents were quickly transferred into pre-warmed 6 mL of OWM and centrifuged (300 x *g*, 5 min, 20 °C). Spheroid pellets were immediately resuspended in 5 mL OWM, washed once, resuspended in a 3:1 mix of Matrigel and OWM, and plated in 50 μL domes. Cryopreserved spheroids were passaged at least once before usage for all analyses.

### Spheroids processing and plating for analysis

All organoid lines used for experiments were between passages 4 and 9. After 5–7 days of passaging with TrypLE Express, Matrigel domes with matured spheroids were dislodged with ice-cold PBS, filtered with a 100 μm strainer (pluriSelect), and centrifuged (300 x *g*, 5mins, 4°C). Pellets were washed once in OWM, manually counted, and replated in domes containing a 1:1 mix of Matrigel and OWM: for 96-well assays, 8 spheroids per μL in 15 μL domes were plated, and for experiments in 24- or 12-well plates, 16 spheroids per μL in 50 μL domes were plated. For all analyses requiring differentiation into colonoids, replated spheroids were allowed to acclimatize overnight in OGM before switching to Intesticult human organoid differentiation medium, ODM (STEMCELL Technologies, #100-0214), for the indicated differentiation durations, and ODM was changed every 2 days. In some

experiments, matured spheroids were passaged and immediately filtered through a 40 μm strainer (Pluriselect) and cultured in OGM for 6–7 days before counting, replating, and differentiation when necessary.

To measure drug-induced responses, 2 μM FCCP (Sigma, #C2920), 50 μM Fenofibrate (Abcam, AB120832), and 1 μM GW6471 (Sigma, #G5045) were added to ODM throughout the differentiation as indicated in figure legends. The solvents served as vehicle controls for respective experiments (DMSO for FCCP, and 100% ethanol for Fenofibrate and GW6471). For inflammatory exposure assays, C and iUC spheroids were exposed to a modified inflammatory cocktail[20] (40 ng/mL TNFα, 20 ng/mL IL-1β, and 500 ng/mL Flagellin) in OGM for 16 hrs, washed with PBS, and differentiated in ODM for 3 days without the cocktail.

### Spheroid Diameter and Colonoid Budding

Spheroids from eight patients per group were passaged and filtered through a 40 μm strainer (Pluriselect). Spheroids were manually counted, and 800 spheroids were replated per 50 μL dome in pre-warmed 12-well plates and cultured in OGM for 6 days. Phase contrast images were obtained on the Keyence BZ-X800 microscope. Morphological measures were conducted by an investigator blinded to the diagnosis. To assess colonoid budding, matured spheroids were counted, replated in 50 μL Matrigel (16 spheroids/μL), and allowed to acclimatize overnight in OGM. Each well was washed with PBS at room temperature (RT) for 3 min before switching to ODM for the indicated number of days. Phase contrast images were obtained on the Keyence BZ-X800 microscope. All measurements of spheroid diameter and enumeration of colonoid budding were performed in a manner blinded to the colonoid diagnosis group using the Keyence BZ-X800 analyzer. The number of buds on each colonoid was presented relative to the total number of visible colonoids per field. The data presented represents an average of 20–30 colonoids per patient sample.

### Extracellular flux analyses

The XF Seahorse MitoStress Test (Agilent) was used to measure oxygen consumption rates (OCR), at baseline, and after the injection of mitochondrial inhibitors[52]. Each spheroid line was replated in quadruplicates in 15 μL matrigel domes (8 spheroids/μL) containing a 1:1 mix of Matrigel and OWM into Seahorse XF96 cell culture plates (Agilent #101085-004) and cultured overnight in 150 μL OGM. The four edges of the plate served as a blank with only a 1:1 mix of Matrigel and OWM. For some experiments, 50 μL of OGM was removed after 24 h into an empty 96-well plate and stored at −80 °C for later measurements of LDH activity. Where differentiation was needed, spheroids were washed once with PBS on the following day and differentiated in 200 μL ODM into colonoids for the indicated number of days in the figure legends, with medium changes every 2 days.

On the day of the assay, ODM was removed from colonoids, washed once with 200 μL PBS, and replaced with 150 μL DMEM assay medium (Agilent, #103334-100), pH 7.4, containing 10 mM glucose, 2 mM glutamine, and 1 mM pyruvate (all from Agilent). Plates were incubated in a non-$CO_2$ incubator for 1 h and OCR was measured at baseline and after a sequential injection of oligomycin (final concentration, 4 μM), FCCP (final concentration, 2 μM), and rotenone/antimycin (final concentration, 100 nM/1 μM) using the XF96$^e$ Extracellular Flux assay kits on the XF96$^e$ analyzer.

The Glycostress test (Agilent) was used to measure extracellular acidification rates (ECAR) in colonoids after a 3-day differentiation. On the day of the assay, ODM was replaced with base medium (Agilent #103575-100) lacking glucose, pyruvate, and glutamine and incubated in a non-$CO_2$ incubator for 1 hr. Baseline ECAR was assessed using the XF96$^e$ analyzer (Agilent) followed by sequential injections of glucose, oligomycin, and 2-deoxy glucose at final concentrations of 40 μM, 4 μM, and 500 mM, respectively. Raw data were normalized with DNA content as described below.

For etomoxir experiments, colonoids were differentiated for 3 days. On the day of the assay, ODM was replaced with assay medium (Agilent #103575) lacking glucose, pyruvate, and glutamine, and incubated in a non-$CO_2$ incubator for 40 min. Etomoxir (Sigma, #E1905) was prepared in the assay medium, quickly added to designated wells within 5 mins at a final concentration of 44 μM, and returned to the non-$CO_2$ incubator for 15 min followed by OCR assessment.

### DNA extraction
For the normalization of colonoid extracellular flux data, DNA normalization protocol was used as previously described with modifications[49]. DNA was extracted using the QIAamp DNA Micro kit (Qiagen #56304). Briefly, the medium was removed from the cell culture plates and 45 μL of ATL lysis buffer was added per well, followed by 5 μL of Protein kinase K. The plate was incubated at 56 °C for 1 h, quadruplicate wells were pooled into 1.5 mL tubes and returned to 56 °C incubator for 1 h. DNA extraction continued according to the manufacturer's instructions. The DNA content of each colonoid line was calculated relative to the blank wells and used to normalize raw data.

### Total ATP measurement
Spheroids were plated in duplicates in clear flat-bottom 96-well tissue culture plates (Falcon #353075) as described for extracellular flux analyses and differentiated for the indicated number of days. Total ATP levels were assayed with the luminescent ATP detection assay kit (Abcam, ab113849), according to the manufacturer's instructions and luminescence was recorded using the Spectramax id5 plate reader (Molecular Devices). ATP levels were normalized by measuring DNA content in a parallel plate.

### Mitochondrial superoxide (MitoSOX) analysis
Spheroids were plated in duplicate in black 96-well tissue culture plates with clear bottoms (Corning, #3603), as described for extracellular flux analyses, and differentiated for the indicated number of days, as specified in the figure legends. Mitochondrial ROS was estimated with the fluorescent probe MitoSOX red (Thermofisher, #M36008). Briefly, a 5 mM MitoSOX stock solution was prepared in DMSO and diluted with HBSS to a working concentration of 3 μM. Colonoids were washed twice with HBSS, and the MitoSOX working solution was added to the colonoids and incubated for 30 min (37 °C, 5% CO2). The colonoids were washed 3 times with HBSS, and MitoSOX fluorescence was measured at 396/610 nm using the Spectramax iD5 plate reader (Molecular Devices). Data were normalized by measuring DNA content as described for extracellular flux analyses.

### Lactate Dehydrogenase (LDH) release
Spheroids were plated in quadruplicates in 96-well plates overnight before commencing differentiation. After 48 hrs, 100 μL of ODM was collected in an empty 96-well plated and stored frozen at −80 °C for later measurements of LDH activity. Extracellular LDH activity was measured using the LDH-Glo Cytotoxicity assay (Promega, #J2380). Samples were diluted 40X in ice-cold storage buffer (200 mM Tris-HCl (pH 7.3), 10% Glycerol, 1% BSA), and LDH activity was measured according to the manufacturer's instructions. Luminescence was recorded using the Spectramax id5 plate reader (Molecular Devices), and the data was normalized with DNA content from the same plate. LDH release was estimated relative to the medium in Matrigel-only blank wells.

### Extracellular glucose and lactate
ODM from colonoids stored from 96 cell culture plates, as previously described for LDH activity, were thawed on ice, and extracellular glucose and lactate were measured using the Glucose-Glo (Promega, #J6021) and Lactate-Glo (Promega, #J5021) assays, respectively. Samples were diluted 50X in ice-cold PBS, and the measurements continued according to the manufacturer's instructions. Luminescence was recorded using the Spectramax ID5 plate reader (Molecular Devices), and the data were normalized with DNA content from the same plate. Glucose consumed or lactate produced was estimated relative to the medium in Matrigel-only blank wells.

### Flow cytometry
**Mitochondrial mass (mtMass) analyses.** For mtMass analyses, we labeled the mitochondria in spheroids and colonoids using the Mito-Tracker Green FM probe (Cell Signaling, #9074). Briefly, single cells from matured spheroids and corresponding differentiated colonoids were obtained by the addition of 900 μL TrypLE Express followed by incubation at 37 °C for 10 mins. Single cells were released by vortexing for 20 seconds with 10-second intervals, and washed in FACS buffer (5% FBS in PBS). Pellets were resuspended in 150 nM mtGreen solution prepared in phenol-red free DMEM/F12 + 5% FBS, followed by incubation for 30 mins at 37 °C in a 5% $CO_2$ incubator. Cells were washed twice in 2 mL PBS and resuspended in 500 μL of ice-cold PBS. The cells were passed through Falcon FACS tubes with a cell strainer cap (Fisher, #0877123) and incubated with 5 μL of the live-dead stain, 7-Aminoactinomycin D (7-AAD), for 5 mins on ice. The mean fluorescence intensity (MFI) of mtGreen was measured in 5000 events using the FITC channel and following 7-AAD exclusion (PE-Cy5) on the Agilent NovoCyte Flow Cytometer. Analysis and images were obtained using the FlowJo software v.10

**Mitochondrial Membrane Potential (MMP).** Single cells from colonoids were processed as described for mtMass analysis and MMP was assessed with the MitoProbe JC-1 assay kit (Thermo Fisher Scientific, MP34152) following the manufacturer's protocol. Briefly, Cells were incubated with 1 μM JC-1 in PBS for 30 mins at 37 °C, 5% $CO_2$. The cells were washed once with PBS and passed through Falcon FACS tubes with a cell strainer cap and incubated with 5 μL of 7-AAD, for 5 mins on ice. 10000 events from each line were recorded using the Agilent NovoCyte Flow Cytometer. After 7-AAD exclusion, MMP was calculated as the MFI ratio of the red (PE)/green (FITC) channels.

**BODIPY neutral lipids.** Matured spheroids from each line were replated in two wells of a 24-well plate and allowed to stabilize overnight in OGM. For each line, one well was switched to ODM to initiate differentiation into colonoids for 2 days while the other well continued in OGM. Single cells were processed from spheroids and colonoids as described for mtMass analysis and cell pellets were fixed using the Cytofix/Cytoperm kit (BD Biosciences, #555028) according to the manufacturer's instructions. Neutral lipid droplets in cells were stained

with 1 µg/mL BODIPY 493/503 (Thermo Fisher Scientific, #D3922) in PBS for 1 h at RT with gentle mixing every 20 mins. Cells were washed once with 2 mL PBS and passed through Falcon FACS tubes with a cell strainer cap. BODIPY MFI was recorded in 5000 events using the FITC channel on the Agilent NovoCyte Flow Cytometer. Analysis and images were obtained using the FlowJo software v.10.

## Immunofluorescence Imaging

**Colonoid embedding in optical cutting temperature (OCT) compound.** Matured spheroids from each line were replated (16 spheroids/µL, 3:1 mixture of Matrigel and OWM) in 6 domes per well of a 6-well plate. After a 3-day differentiation, colonoids were fixed with 5 mL of 2% Paraformaldehyde and 0.1% Glutaraldehyde for 30 min at RT. Colonoids were washed 3X with 5 mL PBS for 10 min, and each Matrigel dome was carefully scooped into a 50 mL tube with 20% sucrose in PBS. After 2–3 days following the sinking of all domes, 6 domes/line was embedded in Tissue Tek OCT compound (Sakura Finetek, #4583) and frozen in liquid $N_2$ using the Seal'N Freeze cryotray and box (Fisher Scientific, #NC1877501). From the Colonoid OCT block, 10-µm-thick cryosections were cut onto slides and stored at −80 °C. For UCP2 and COX4 immunostaining, cryosections were washed once with PBS for 15 secs at RT followed by autofluorescence quenching by incubation with 10 mM Sodium Borohydride in PBS twice for 5 mins at RT (0.2 M Glycine was used for other markers). Sections for UCP2 and COX4 were washed thrice in PBS for 10 minutes each, followed by permeabilization with 0.15% Triton X-100 for 15 min at RT. Slides were washed thrice with PBS and incubated with the blocking buffer−3% BSA, 5% donkey serum, and 5% goat serum in PBS (10% FBS in PBS for other markers) for 1 hr at RT. This was followed by overnight incubation with primary antibodies anti-UCP2 (1:100, Proteintech, #11081-1-AP), anti-COX4 (1:200, Invitrogen, GT6310), anti-AVIL (1:100, Sigma-Aldrich, HPA058864), anti-OLFM4 (1:140, Abcam, ab85046), anti-SLC26A3 (1:100, Novus Biologicals, NBP1-84450) and mouse anti-MUC2 (1:200, Invitrogen, MA5-12345) in the blocking buffer in a humidified chamber at 4 °C. Slides were washed twice with PBS for 15 secs, followed by 2X wash in PBS for 5 mins. Slides were incubated in a humidified chamber for 1 hr at RT in secondary antibodies: donkey anti-rabbit Alexa Fluor 647 (1:1000, Invitrogen, #A31573), goat anti-mouse Alexa Fluor 488 (1:1000, Invitrogen, #A11001), goat anti-rabbit Alexa Fluor 594 (1:100, Invitrogen, #A37117), and goat anti-rabbit Alexa Fluor 488 (1:400, Invitrogen, #A32766) prepared in their respective blocking buffer. Slides were washed 3X in PBS and counterstained with Phalloidin (1:250, ThermoFisher Scientific #T7471) in PBS for 30 min at RT. Slides were washed once in distilled water and mounted in Prolong Glass Antifade Mountant with NucBlue Stain containing the Hoechst 33342 DNA marker (ThermoFisher Scientific, #P36983). Images were acquired on the Keyence BZ-X800 microscope. Images were merged with the Keyence BZ-X800 software analyzer and cells expressing DAPI and those co-expressing UCP2 and COX4 were counted on at least four high-power fields (60X oil objective) per patient colonoid line. For other markers, each marker was identified and counted on four organoids (40X objective) per patient colonoid line in a blinded evaluation manner. Total cell counts per colonoid were counted using the hybrid cell count function in the Keyence BZ-X800 software analyzer with uniform settings.

**Colonoid whole-mount staining.** For whole-mount staining, matured spheroids were replated in 8-well chamber µ-slides (IBIDI, #80826). After differentiation, colonoids were washed with PBS at RT and fixed with 500 µL of 4% PFA for 30 mins at RT. For UCP2 staining, colonoids were washed 2X with 500 µL PBS for 5 mins and permeabilized with 0.2% Triton X-100 in PBS for 30 mins at RT followed by 2X wash in PBS. Samples were incubated with 50 mM ammonium chloride ($NH_4Cl$) in PBS for 30 mins at RT to quench potential autofluorescence. After a 2X wash in PBS, samples were incubated in blocking buffer (3% BSA, 5%

donkey serum in PBS) for 1 hr at RT and incubated in primary antibody-rabbit anti-UCP2 (1:50, Proteintech, #11081-1-AP) overnight and secondary antibody- donkey anti-rabbit Alexa Flour 647 (1:500, Invitrogen, #A31573) for 1 hr at RT, and incubated with 300 µL Phalloidin prepared as described above. Samples were mounted in 200 µL Fructose-glycerol clearing solution[53] and incubated for 30 mins at RT. Whole-mount images were acquired using the Stellaris 8 Inverted Confocal Microscope (Leica) using the 20x oil objective and individual channels were merged using the Leica LAS X software (v. 5.0.2).

For visualizing neutral lipids, colonoids differentiated for 2 days in chamber slides were fixed in 4% PFA, washed, and incubated in 50 mM $NH_4Cl$ as described above. Then colonoids were stained with 1 µg/mL BODIPY 493/503 (Thermo Fisher Scientific, #D3922) in PBS for 1 h at RT, washed twice in PBS, and incubated with Phalloidin as described above. Chamber slides were washed once in distilled water and mounted in 2 drops of Prolong Glass Antifade Mountant with NucBlue Stain containing the Hoechst 33342 DNA marker (ThermoFisher Scientific, #P36983). Samples were allowed to cure overnight at RT in the dark. Whole-mount images were acquired using the Stellaris 8 Inverted Confocal Microscope (Leica).

## RNA processing and Bulk RNA sequencing

RNA was processed concurrently from eight patient organoid lines per diagnosis using the RNeasy Micro Kit (Qiagen, #74004). Briefly, matured spheroids from each line (C, aUC, iUC, $n = 8$ per group) were filtered through a 100 µm strainer (Pluriselect), replated (16 spheroids/µL, 2:1 mixture of Matrigel and OWM, 50 µL/dome) in two wells of a 12-well plate (3 domes/well) and allowed to acclimatize overnight in 800 µL OGM. The following day, one well of spheroids/line was processed for RNA extraction, while the other well was switched to ODM for differentiation into colonoids for 3 days. For RNA extraction, the medium was removed, and Matrigel domes were dislodged with PBS from the bottom of the plate. To digest and remove any traces of Matrigel, the suspension was centrifuged (300 x g, 5 mins, 4 °C), 350 µL of TrypLE Express (Gibco) was added to the spheroid pellets and incubated in a 37 °C water bath for 3 mins. The tube was swirled briefly followed by the addition of 5 mL ice-cold OWM. Samples were washed (300 x g, 5 mins, 4 °C), and the supernatant was aspirated. Any remnants of OWM were carefully removed with a 200 µL pipette and the resulting pellets were resuspended in the lysis buffer from the RNeasy Micro Kit and stored immediately at −80 °C. Further processing of RNA followed the manufacturer's instructions. RNA sequencing of poly-A enriched RNA libraries was carried out on the Illumina NovaSeq PE150 platform (Novogene). RNA-seq reads were aligned to the human reference hg38 genome and the nf-core/rnaseq (v3.5) workflow was used for developing the expression matrix.

## Ingenuity pathway analysis (IPA)

Fold changes from the colonoid bulk RNA-seq data (Log2FC and p-adj values) were uploaded into the IPA software and used to predict canonical pathways that underlie transcriptional data. All upregulated and downregulated genes were concurrently analyzed in IPA with an expression p-value cutoff of 0.1. The top biological and cellular pathways and causal networks were selected based on the fisher's exact test p-value which measures the overlap of observed and predicted regulatory gene sets[27].

## Single cell data analyses

We utilized data from the established single-cell transcriptomic atlas of 68 colon biopsies obtained from 18 UC patients and 12 healthy controls[29]. Processed single-cell expression matrices and metadata for human colon epithelial cells were obtained from the Single Cell Portal (https://singlecell.broadinstitute.org/single_cell, accession SCP259). Expression counts were normalized using the R

(v.4.3.3) package Seurat (v.5.1.0). The reported PPARA, RXR, and lipid-related genes were visualized using the R package ComplexHeatMap (v.2.18.0).

## Quantitative RT PCR

For most analyses, total RNA was processed from spheroids or colonoids by matrigel removal and digestion as described above for RNA processing. Due to the short half-life of *UCP2*[54], samples for *UCP2* analysis were plated in a 1:1 mixture of Matrigel and OWM, differentiated, and lysed directly in the culture well without Matrigel removal. For all analyses, RNA was processed (RNeasy Micro Kit, Qiagen, #74004) and cDNA was synthesized using the High-Capacity cDNA Reverse Transcription kit (Applied Biosystems, #4368814). Gene expression was quantified on the QuantStudio 3 thermocycler using Taqman probes from Invitrogen (Supplementary Table 2). Fold changes were quantified using the $2^{-\Delta\Delta CT}$ method using *ACTB* or *GAPDH* as the reference gene where appropriate.

## Lipidomics

Colonoids were differentiated for 3 days and digested into single cells as previously described above for flow cytometry. Single cells were counted, and lipids were extracted in $0.5 \times 10^6$ cells using 1:1 butanol/methanol with 10 mM ammonium formate, vortexed and bathsonicated for 1 h at 21–25 °C. Following sonication, samples were centrifuged (13,000x *g*, 10 mins, 20 °C), and the supernatant was transferred to the glass vial and spiked with Avanti's Splash II Lipidomix mass spec standard as internal standards. dMRM LC/MS analysis was conducted on an Agilent 1290 Bio LC system coupled to the 6495 C TQ MS. Separation was achieved using a ZORBAX Eclipse Plus C18 column (2.1 ×100 mm, 1.8 μm) with mobile phases: (A) 10 mM ammonium formate, 5 μM deactivator additive in a mixture of 5:3:2 water:acetonitrile:2-propanol, and (B) 10 mM ammonium formate in a mixture of 1:9:90 water:acetonitrile:2-propanol. This targeted lipidomics assay provides a broad coverage of 762 lipid species across 44 lipid classes. Each sample was injected twice, and 739 lipids were detected from the samples. Data was analyzed using MassHunter Quant software (v. 12.1) and imported to Agilent Mass Profiler Professional (v. 15.1) for statistical analysis. A full description of LC/MS method and data analysis is available in the Supplementary Information.

## LPL Activity

Spheroids were plated, differentiated for 2 days or 5 days, and processed for Matrigel removal as described above for RNA processing. Total colonoid protein was isolated using the Cell Lytic buffer (Sigma, #2978) with 1X Protease and Phosphatase inhibitor cocktail (ThermoFisher Scientific, #78440). Colonoids were incubated on ice for 30 mins with brief vortexing every 10 mins. Samples were centrifuged (10000 x *g*, 10 mins, 4 °C), and the supernatant was transferred into fresh 1.5 mL tubes. Protein concentration was determined using the BCA protein Assay kit (ThermoFisher Scientific, #23227). LPL activity in each sample was determined in duplicates using the LPL activity assay kit (Cell Biolabs, #STA-610) following the manufacturer's instructions. Readings were obtained in a Spectramax id5 plate reader (Molecular Devices), and the data was normalized to total protein concentration.

## Nuclear PPAR-α activity assay

Three wells of spheroids (3 domes/well) per donor were plated in 12-well plates, differentiated in ODM for 24 hrs, pooled, and processed for Matrigel removal. Nuclei isolation followed the established Omni-ATAC protocol used for nuclei preparations for ATAC-seq[55]. Nuclei protein concentration was determined using the BCA protein Assay kit (ThermoFisher Scientific, #23227). PPAR-α activity was assessed in the nuclear extracts with the PPAR-α activity kit (Abcam, #ab133107) following the manufacturer's instructions.

## Extracellular chemokine assay

Spheroids were plated and differentiated in ODM for the indicated days in the figure legend, as described above for RNA processing. ODM was collected at the indicated days and frozen at −80 °C until analyses. For analyses involving differentiation until day 5, ODM was changed on day 2 before collection and analyses on day 5. The total colonoid protein was determined using the Cell Lytic buffer with 1X Protease and Phosphatase inhibitor. Extracellular chemokines in ODM on the indicated days were assessed in duplicates using a Spectramax id5 plate reader (Molecular Devices) and the ELISA kits (CXCL1: Abcam #ab190805, CXCL11: ThermoFisher #EHCXCL11, CCL2: ThermoFisher #88-7399-22) following manufacturer's instructions.

## Neutrophil chemotaxis assay

Spheroids were replated in 50 μL/dome (20 spheroids/μL) and differentiated in ODM for 2 days. Thereafter, colonoids were then switched to protein-free DMEM/F12 (Corning, #10-092-CV) supplemented with 50 μg/mL Normocin. After 24 h, the conditioned media was collected and used for neutrophil migration assay. The colonoids were processed for total protein determination.

Neutrophils were isolated from 10 ml whole blood of a healthy adult using MACSxpress® Whole Blood Neutrophil Isolation Kit (Militenyi Biotec #130-104-434) following manufacturer's instructions. Neutrophils were resuspended in RPMI 1640 and a chemotaxis assay was done using a Boyden chamber in the cell migration kit (Abcam #ab235696) following the manufacturer's instructions. Briefly, $3.5 \times 10^5$ cells were added to the top of a 24-well transwell with inserts of 3-μm pore size (Corning, 3415). A total of 600 μl of condition media from each colonoid line was added to the bottom of the transwell and the cells were kept at 37 °C, 5% $CO_2$ for 2 hours. 500 ng/mL CXCL1 recombinant protein and the inducer from manufacturer's kit were used as a positive control (ThermoFisher #300-11-1MG). A standard curve of neutrophil number was prepared in a 96-well plate according to the manufacturer's instructions. After 2 hours, neutrophils in the top insert were removed and wiped by cotton swaps. Invasive cells in the bottom chamber were stained with the cell dissociation solution/cell dye and 110 μL of mixture from each colonoid line were transferred in triplicates into the same 96 well plates with the standard and fluorescence were measured using a Spectramax id5 plate reader (Molecular Devices). The number of invasive neutrophils was calculated using the standard curve and normalized by total protein concentration from each colonoid line.

## Statistics

Eight patient-derived colonoids were generated per diagnosis (control, active UC, inactive UC). For experiments comparing metabolic measures between diagnoses, a sample size of 8 patient colonoid lines per group was chosen to provide 80% power to detect a difference of 1.5–1.8 times the standard deviation between groups. Based on initial results from metabolic studies and RNA-seq, and with inhibitor studies, larger effect sizes were anticipated, and a minimum of n 3 per group was used. Symbols in figures represent the average measures of individual patient colonoid lines used for each assay as described in the figure legends. Wherever representative colonoids were used, the number of independent colonoids and technical replicates used were described in the relevant figure legend or method section. Statistical tests were summarized in each figure legend. Analysis of two groups was conducted with a two-sided Mann-Whitney test or unpaired T-test, and Welch correction was done for data with unequal variance. Wherever colonoid lines from the same diagnosis were treated and compared, a two-sided paired T-test or Wilcoxon signed-rank test was used. Comparisons of more than two groups were done with one-way or two-way ANOVA with Tukey or Holm-Sidak test. When present, error bars indicate mean ± SE. Box-plot elements generally include boxplot limits within the interquartile range, center line as median, with

whiskers to min and max points unless otherwise stated in the figure legend. Statistical analysis was conducted in Prism (v.10.0), and *P* values < 0.05 were considered statistically significant. For differentially expressed genes from the RNA-seq data, transcripts with $Log_2FC \geq \pm 1$, FDR $P \leq 0.1$ were considered statistically significant.

## Reporting summary

Further information on research design is available in the Nature Portfolio Reporting Summary linked to this article.

## Data availability

All data supporting the findings in this study are available either within the paper, its Supplementary Information, Source data file, and appropriate data repositories. The bulk RNA sequencing data in this study have been deposited into Gene Expression Omnibus (GEO) under the accession number GSE276170. This study used the publicly available bulk RNA sequencing datasets under the accession numbers GSE109142 and GSE117993. This study also utilized publicly available single cell RNA sequencing data from the Single Cell Portal under the accession number SCP259. Lipidomics data are provided in Supplementary Information. Source data are provided with this paper.

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

## Acknowledgements

We thank Lance S. Prince for the resources for flow cytometry and confocal microscopy. Research reported in this publication was supported by the National Institute Of Diabetes And Digestive And Kidney Diseases of the National Institutes of Health (NIH) under Award Numbers R21DK123691 (to M.J.R), K99DK136971 to B.A.O, and R01DKDK099222 to S.D.; the National Heart, Lung, and Blood Institute of the NIH under Award Numbers R01HL139664, R01HL160018, R01 HL134776, R01HL59886, 1R01HL172449-0 to V. D.J.P.; and the National Center for Advancing Translational Sciences of the NIH under Award Numbers UM1TR004921 and UL1TR003142. The content is solely the responsibility of the authors and does not necessarily represent the official views of the NIH. Research was also supported by the Kenneth Rainin Foundation through an Innovator Award 2024-0046 to M.J.R. Additional support was provided by the Stanford Medicine Children's Health Center for IBD and Celiac Disease (fellowship award to B.A.O.) and the Stanford Innovative Medicines Accelerator (to MJR). This work utilized computing resources provided by the Stanford Genetics Bioinformatics Service Center and metabolomics resources through Stanford Sarafen ChEM-H.

## Author contributions

Conceptualization: B.A.O., S.D., and M.J.R., Organoid biobank generation: B.A.O., Y.Z., S.R.F., A.W., M.G., T.T., A.L.D., and M.J.R., Methodology: B.A.O., Y.Z., S.R.F., Y.Q., M.E.M., S.D., Y.D., V.D.J.P., and M.J.R, Investigation: B.A.O., Y.Z., L.H., M.G., Y.Q., E.I.A.E., N.S., and Y.D., Analysis and Visualization: B.A.O., Y.Z., L.H., J.A.L., and Y.D., Project administration: B.A.O., A.L.D., V.D.J.P., and M.J.R, Supervision: M.J.R., Writing – original draft: B.A.O., Writing – review & editing: B.A.O., M.J.R., Manuscript approval: All authors, Funding acquisition: S.D., A.L.D., and M.J.R.

## Competing interests

The authors declare no competing interests.
