## [Transparent Peer Review file · Nature Communications]

Patient-derived colon epithelial organoids reveal lipid-related metabolic dysfunction in pediatric ulcerative colitis

Corresponding Author: Dr Michael Rosen

Version 0:

Reviewer comments:

Reviewer #1

(Remarks to the Author)

This study utilizes human patient-derived colonoids from normal and ulcerative colitis to demonstrate that lipid accumulation and metabolism are altered in ulcerative colitis epithelium. The authors further show a role of PPAR α in driving these alterations. Strengths of the study include the translational approach and implications, an intriguing link between metabolic stress and epithelial-produced chemokine CXCL1 suggesting a novel mechanism of immune cell regulation/recruitment relevant to IBD, unbiased transcriptomics, and mechanistic studies using PPAR α antagonist. This is a well-executed and interesting study. There are a few weaknesses including the need to better demonstrate inability of aUC colonoids to differentiate into any epithelial lineage, no demonstration of functional immune change driven by increased chemokine expression (that could be considered a slight increase), and a few other specific comments:

1. Is there an inability of the aUC colonoids to differentiate into any and all mature epithelial lineages (since CHA is decreased, and WFDC2 is decreased)? Please also include an absorptive lineage marker or colonocyte marker and stain colonoids to identify whether enteroendocrine cells, goblet cells, and colonocytes develop in aUC colonoids.
2. An alternate reason aUC exhibit increased OCR and mitoROS (figure 1) is that their composition is skewed toward predominance of colonocytes, which have been indicated to exhibit higher mitochondrial respiration compared to other differentiated epithelial cells. The authors should rule this out (related to comment 1).
3. Can the authors functionally show the increased chemokine expression in aUC actually drives chemoattraction of immune cells?
4. Line 253 states that Eto reduced proton leak in aUC but not control colonoids. Although it is marked not significant in Figure 4H, it seems Eto decreases proton leak in control colonoids ~50%, which seems to be a similar decrease to that in aUC. Therefore, it is hard to conclude that FAO drives proton leak specifically in aUC. Perhaps this should be tested in more than 3 donors/group. This is especially important given the range of proton leak across donors shown in Fig 1F, where 4 aUC donors exceed control values and 4 aUC donors fall in the same range as control values - it would be useful to know where the 3 donors in Fig 3H fall in this range noted in Fig 1F.
5. In regards to Figure 4L-O, the idea that lipids enhance stemness and inhibit differentiation in colonoids is well-established. However, it is novel to link this to chemokine expression in the epithelium. Unfortunately, the increase in CXCL1 mRNA in Fig 4O is slight. Does this equate to increased CXCL1 protein secretion or actual immune cell recruitment? Wondering if this ~1.4 fold increase in CXCL1 mRNA actually drives a functional response, which would be important to demonstrate.
6. Figure 4O, is this induction of CXCL1 by BSA-PA dependent on FAO? For instance, if Etomoxir is included during BSA-PA would the increase in CXCL1 be prevented?
7. Although the authors show dysregulated lipid metabolism genes from PROTECT study in active UC (Supp Fig 5) and that BODIPY⁺ neutral lipids are increased in aUC colonoids (Fig 4B), lipidomics on control and aUC colonoids could be more informative on lipid type. BODIPY staining will include many lipids (fatty acids, cholesterol, ceramides etc) and it would be quite interesting if specifically fatty acids used for FAO are increased in aUC.

8. Minor: Are the labels correct on the X axis for supplementary fig 4B and C? This seems to be data from RISK and PROTECT studies but labels suggest it to be control and aUC colonoids - please verify

9. Minor: Line 324, this should be Figure 6L not 6I

Reviewer #2

(Remarks to the Author)

The current manuscript by Ojo et al. demonstrates in a pediatric patient-derived colonoid model that hypermetabolism and metabolic dysregulation of lipids contribute to epithelial damage in ulcerative colitis via PPAR-alpha signaling. Published datasets are leveraged for validation. Notably, PPAR-alpha agonist recapitulated in control colonoids the phenotype observed in aUC colonoids, and pharmacological inhibition of PPAR-alpha restored lipid metabolism in aUC colonoids (albeit at the cost of increased glucose consumption). The study provides important mechanistic insight into metabolic abnormalities within colonic epithelium of patients with UC, as well as a potential approach for leveraging these findings therapeutically. However, there are several concerns in the current manuscript that should be addressed:

1. The sex distribution varies in the patient cohorts: control cohort is 2% male, iUC 88% male, aUC 50% male. While recruitment of pediatric subjects is challenging and it may not be feasible to aim for an even distribution, this issue should be mentioned in the main text.
2. While the hypermetabolic phenotype of colonic epithelium in aUC and its consequences are well defined in the study, it is unclear what induces the metabolic uncoupling in vivo in the first place. While addressing this experimentally is outside of the scope of this study, the Discussion would benefit from a hypothesis.
3. Fccp treatment within a 3-day differentiation span induced metabolic changes without the morphological changes in colonoids. Could chronic exposure be needed to impact differentiation?
4. While PPARa activation resulted in profound changes in control colonoids, PPARa inhibition rescued metabolic features but not gross morphology of aUC colonoids – please address this in Discussion.
5. Fig 4L: the change in gene expression is statistically significant, but weak. In addition, colonoids from only one donor have been used in these experiments. Each symbol on the plots represents one well of culture, which is a suboptimal technical replicate. If it wasn't feasible to use biological replicates (colonoids from different donors), results from independent passages of colonoids from one donor should be reported.
6. Can the phenotypic differences presented in Fig 4N be quantified?
7. P13 line 298: please rephrase “overexpressed PPAR-alpha activity”, as activity is not something that is expressed.
8. Fig 6D is not referenced in Results
9. P44 line 950: the phrase in legend for Figure 5C “... treated as in C” likely should state “...treated as in B”.

Technical concerns:

- Please report colonoid passage range for each experiment (only provided for RNA-seq in the current manuscript).
- Please report seeding density for colonoids (i.e. the number of crypts seeded per Matrigel dome).
- In experiments with re-exposure to inflammation, was the inflammatory cocktail only added for 16 h prior to differentiation or did the treatment continue throughout differentiation as well?
- P30 line 703: “Three spheroids per group were differentiated...” – perhaps it was three domes or wells of spheroids per group?
- The inflammatory cocktail was modified from the original study by Arnauts et al to have lower concentration of TNFalpha. It would be helpful to have the reason for this modification clarified.
- P19 line 442: please confirm that spheroid expansion was carried out in pure (not 50%) L-WRN conditioned medium with no supplements (i.e. nicotinamide, B-27, etc.)

Version 1:

Reviewer comments:

Reviewer #1

(Remarks to the Author)

The authors have adequately addressed my previous concerns and have provided an exciting study

Reviewer #2

(Remarks to the Author)

The authors addressed all of my concerns. There are some minor errors in the manuscript that should be fixed:

Figure 2: a fragment of transwell is visible at the bottom of panel G

Supplementary Fig. 4:

- Panel E overlapping with a schematic showing sulfite addition prior to differentiation – perhaps this is misplaced
- Panel J shows two identical Basal Respiration box plots overlapping each other

RESPONSE TO REVIEWERS

We thank the reviewers for their thoughtful and constructive feedback, which has significantly improved the quality of our manuscript. In response, we have conducted additional experiments to address the key concerns raised, including characterization of lipids in the colonoids via targeted lipidomics, chemotaxis assay, longer Fccp treatments for its impact on differentiation, and extracellular protein evidence for BSA-PA's impact on chemokine expression. We have also revised the manuscript to include the new data and strengthened the interpretation of results with the hope that these additions will make our manuscript suitable for publication. Below, we provide a point-by-point response to each comment.

Reviewer Comments

Reviewer #1 (Remarks to the Author):

This study utilizes human patient-derived colonoids from normal and ulcerative colitis to demonstrate that lipid accumulation and metabolism are altered in ulcerative colitis epithelium. The authors further show a role of PPARa in driving these alterations. Strengths of the study include the translational approach and implications, an intriguing link between metabolic stress and epithelial-produced chemokine CXCL1 suggesting a novel mechanism of immune cell regulation/recruitment relevant to IBD, unbiased transcriptomics, and mechanistic studies using PPARa antagonist. This is a well-executed and interesting study. There are a few weaknesses including the need to better demonstrate inability of aUC colonoids to differentiate into any epithelial lineage, no demonstration of functional immune change driven by increased chemokine expression (that could be considered a slight increase), and a few other specific comments:

1. Is there an inability of the aUC colonoids to differentiate into any and all mature epithelial lineages (since CHA is decreased, and WFDC2 is decreased)? Please also include an absorptive lineage marker or colonocyte marker and stain colonoids to identify whether enteroendocrine cells, goblet cells, and colonocytes develop in aUC colonoids.

Response: We thank the reviewer for the comment. We differentiated aUC and C colonoids for 3 days and performed immunofluorescent staining for stem/progenitor and mature epithelial lineages. We demonstrated an inability of the aUC colonoids to differentiate into goblet and mature colonocytes, but not tuft cells. We found no differences in the stem/progenitor cell marker OLFM4 and tuft cells marker Advillin in aUC compared to C. However, we found reduced goblet cell maker, MUC2 and mature colonocyte marker SLC26A3 in aUC compared to C colonoids (now, Supplementary Fig 3A-D and Line 107-110). We did not observe robust enteroendocrine signals with our CHGA staining at this stage of differentiation suggesting a need for longer differentiation to observe this.

2. An alternate reason aUC exhibit increased OCR and mitoROS (figure 1) is that their composition is skewed toward predominance of colonocytes, which have been indicated to exhibit higher mitochondrial respiration compared to other differentiated epithelial cells. The authors should rule this out (related to comment 1).

Response: We appreciate the reviewer for the comment. As we now show and as indicated in the response to the comment above, we stained for the chloride transporter, SLC26A3 and we observed a reduction in aUC colonoids vs C, similar to a previous report also in human colonoids ⁽¹⁾. This suggests that the metabolic phenotypes we observed at this stage of differentiation may not be explained by the dominance of colonocytes.

3. Can the authors functionally show the increased chemokine expression in aUC actually drives chemoattraction of immune cells?

Response:

We performed a neutrophil chemotaxis assay and demonstrated that the medium from aUC colonoids recruited more neutrophils than that from C colonoids. Thus, we confirmed that the increased chemokines expressed by differentiated aUC colonoids are functionally significant. We had to modify our differentiation protocol slightly to account for the fact that organoid differentiation media (ODM) contains serum, which has strong chemoattractive potential. Therefore, for the final 24 hours before harvesting conditioned media, the organoids were incubated in serum-free DMEM/F12. We confirmed by ELISA that the aUC organoid conditioned media under this modified protocol also contained increased CXCL1 protein (Response Fig. 2 below). The chemotaxis assay has been added to Fig. 2., results (Line 204-211), and methods (Line 791-809)

CXCL1 concentration in conditioned medium for neutrophil chemotaxis assay

Response Figure 1: CXCL1 concentration in conditioned medium for neutrophil chemotaxis assay

4. Line 253 states that Eto reduced proton leak in aUC but not control colonoids. Although it is

marked not significant in Figure 4H, it seems Eto decreases proton leak in control colonoids ~50%, which seems to be a similar decrease to that in aUC. Therefore, it is hard to conclude that FAO drives proton leak specifically in aUC. Perhaps this should be tested in more than 3 donors/group. This is especially important given the range of proton leak across donors shown in Fig 1F, where 4 aUC donors exceed control values and 4 aUC donors fall in the same range as control values - it would be useful to know where the 3 donors in Fig 3H fall in this range noted in Fig 1F.

Response:

We have indicated the samples in Fig 1F used for the Eto analysis (Fig 4I) in Response Figure 3 below. We have also rephrased the language to acknowledge the reduction in C (Line 278-280). The absolute levels may have been different due to different media conditions. On average, there was a 55% reduction in C and 65% reduction in aUC. This experiment in a nutrient-depleted condition forces the organoids to use their substrate stores to sustain their metabolic activity for the 3 hr period of the mitostress assay (1 hr pre-assay incubation and ~2 hr of the assay). Lipid-induced metabolic activity is known to contribute to mitochondrial uncoupling⁽²⁻⁴⁾. Thus, some Eto-induced reduction in proton leak in C colonoids is not surprising since C colonoids depended mildly on lipid stores during this nutrient-scarce period for basal respiration, although at much lower rates than aUC (Fig 4H). Overall, this assay showed that aUC colonoids have a higher potential for lipid-induced respiration, which is supported by the increased lipid content, particularly the enhanced acylated lipids in the lipidomics data.

Response Figure 2: Colonoid donors from Fig 1F used for etomoxir assays in Fig 4G-J

5. In regards to Figure 4L-O, the idea that lipids enhance stemness and inhibit differentiation in colonoids is well-established. However, it is novel to link this to chemokine expression in the epithelium. Unfortunately, the increase in CXCL1 mRNA in Fig 4O is slight. Does this equate to increased CXCL1 protein secretion or actual immune cell recruitment? Wondering if this ~1.4 fold increase in CXCL1 mRNA actually drives a functional response, which would be important to demonstrate.

Response: We measured protein secretion, and it confirmed a modest but uniform increased protein production that normalized with etomoxir (see Fig 4M-N, Line 298-303 in manuscript).

6. Figure 4O, is this induction of CXCL1 by BSA-PA dependent on FAO? For instance, if Etomoxir is included during BSA-PA would the increase in CXCL1 be prevented?

Response: We thank the reviewer for the vital question. Yes, we conducted this experiment and found that CXCL1 secretion was dependent on FAO. (see Figure 4M,N in manuscript and discussed in Line 300-303 in manuscript)

7. Although the authors show dysregulated lipid metabolism genes from PROTECT study in active UC (Supp Fig 5) and that BODIPY+ neutral lipids are increased in aUC colonoids (Fig 4B), lipidomics on control and aUC colonoids could be more informative on lipid type. BODIPY staining will include many lipids (fatty acids, cholesterol, ceramides etc) and it would be quite interesting if specifically fatty acids used for FAO are increased in aUC.

Response: We thank the reviewer for this suggestion. We conducted a lipidomics analysis and found several ceramides and 20 overabundant acyl-FAs in aUC, many of which may be used to support FAO. Interestingly, all significant lipids classes observed are overabundant in aUC compared to C. We have added the lipidomics results summary to Figure 4 E and F, and discussed in Line 265-271. We also submitted the full data as Supplementary Data 3.

8. Minor: Are the labels correct on the X axis for supplementary fig 4B and C? This seems to be data from RISK and PROTECT studies, but labels suggest it to be control and aUC colonoids - please verify

Response: We thank the reviewer for their comment. We agree and this has been changed to UC instead of aUC to distinguish it from the colonoid labels.

9. Minor: Line 324, this should be Figure 6L not 6I

Response: We appreciate the reviewer for the correction. This has been edited in Line 354.

Reviewer #2 (Remarks to the Author):

The current manuscript by Ojo et al. demonstrates in a pediatric patient-derived colonoid model that hypermetabolism and metabolic dysregulation of lipids contribute to epithelial damage in ulcerative colitis via PPAR-alpha signaling. Published datasets are leveraged for validation. Notably, PPAR-alpha agonist recapitulated in control colonoids the phenotype observed in aUC colonoids, and pharmacological inhibition of PPAR-alpha restored lipid metabolism in aUC

colonoids (albeit at the cost of increased glucose consumption). The study provides important mechanistic insight into metabolic abnormalities within colonic epithelium of patients with UC, as well as a potential approach for leveraging these findings therapeutically. However, there are several concerns in the current manuscript that should be addressed:

1. The sex distribution varies in the patient cohorts: control cohort is 2% male, iUC 88% male, aUC 50% male. While recruitment of pediatric subjects is challenging and it may not be feasible to aim for an even distribution, this issue should be mentioned in the main text.

Response: We acknowledge that the control cohort is 25% male, the iUC cohort 88% male, and aUC cohort 50% male. To assess whether this imbalance may have confounded disease status with regard to metabolic differences, we performed a stratified analysis comparing the basal oxygen consumption rates between male and female patients within each diagnosis. Both male and female patient organoids still showed statistically significant differences in basal OCR between aUC and C. We have included a summary of this analysis in the revised manuscript (Supplementary Fig. 3G and Result section in Line 124-126).

2. While the hypermetabolic phenotype of colonic epithelium in aUC and its consequences are well defined in the study, it is unclear what induces the metabolic uncoupling in vivo in the first place. While addressing this experimentally is outside of the scope of this study, the Discussion would benefit from a hypothesis.

Response: We appreciate the reviewer's suggestion. In response we have added the following statement regarding our hypothesis to the discussion (Line 416 – 425):

“Natural PPAR- α agonists including PUFAs are disproportionally accumulated in the inflamed colon mucosa of UC patients which correlated positively with endoscopic disease activity.^{41,42} Sphingolipids may induce lipid oxidation via PPAR- α ⁴³ and bacterial-derived sphingolipids enhance susceptibility to DSS-induced colitis⁴⁴. Our colonoid lipidomics analysis revealed an overaccumulation of 2 PUFAs, 7 acylated PUFAs and 26 sphingolipid species in aUC. Considering that lipids are known to induce metabolic uncoupling in other tissues⁴⁵⁻⁴⁷ similar to the outcomes in our study, we hypothesize that the overexposure of colon epithelial stem cells to dietary or bacterial-derived LCFAs, PUFAs or complex lipids in vivo will promote epithelial hypermetabolic phenotypes, uncoupling and epithelial metabolic inefficiency during colon epithelial differentiation. Whether this happens in a PPAR- α dependent manner in vivo could be the subject of future studies.”

3. Fccp treatment within a 3-day differentiation span induced metabolic changes without the morphological changes in colonoids. Could chronic exposure be needed to impact differentiation?

Response: We thank the reviewer for their comment. We also thought this was an interesting question to ask. We found that exposure of control colonoids to FCCP for 6 days significantly reduced colonoid budding and decreased *WFDC2* expression. This suggests that metabolic dysregulation is an initial process that precedes visible morphological changes in differentiation. We have added this to Supplemental Fig. 4 P-S, Results (Line 178-180).

4. While PPAR α activation resulted in profound changes in control colonoids, PPAR α inhibition rescued metabolic features but not gross morphology of aUC colonoids – please address this in Discussion.

Response: We thank the reviewer for their comment. There may be other dysregulated genes and pathways in aUC that independently impact symmetry breaking and bud development during differentiation. We continue to explore other possibilities around this subject in future studies. We have added a discussion on this in Line 434-438.

5. Fig 4L: the change in gene expression is statistically significant, but weak. In addition, colonoids from only one donor have been used in these experiments. Each symbol on the plots represents one well of culture, which is a suboptimal technical replicate. If it wasn't feasible to use biological replicates (colonoids from different donors), results from independent passages of colonoids from one donor should be reported.

Response: We initially conducted this short (24 hr) experiment to test the minimal PA dose that impacts genes related to lipid transport (FABP6) and uncoupling (UCP2) for subsequent experiments (now Supplemental Fig. 6C-D). We have tested this dose in 4 independent donor lines and we still see that slight but significant effects in only FABP6 and UCP2 in all lines but WFDC2 expression was not significant. This data is now in Supplemental Fig. 6C-D.

6. Can the phenotypic differences presented in Fig 4N be quantified?

Response: We thank the reviewer for the suggestion. We have now quantified the BSA-PA effect on morphology and confirmed reduction in colonoid budding (now Supplemental Fig. 6F).

7. P13 line 298: please rephrase “overexpressed PPAR-alpha activity”, as activity is not something that is expressed.

Response: We thank the reviewer for pointing out the error. We have changed this to “enhanced PPAR- α activity”. Now Line 315

8. Fig 6D is not referenced in Results

Response: We thank the reviewer for pointing out the error. We have referenced this in the Results (Line 338).

9. P44 line 950: the phrase in legend for Figure 5C “... treated as in C” likely should state “...treated as in B”.

Response: We thank the reviewer for pointing out the error. We have corrected this in the figure legend (now Line 1065).

Technical concerns:

- Please report colonoid passage range for each experiment (only provided for RNA-seq in the current manuscript).

Response: We appreciate the reviewer's comment. All experiments used colonoids between passages 4 and 9. We have reported this in the Methods (Line 507)

- Please report seeding density for colonoids (i.e. the number of crypts seeded per Matrigel dome).

Response: We have clarified this in the Methods (Line 509-512). For 96-well assays, we seeded 8 spheroids per μL in 15 μL domes and for experiments in 24- or 12-well plates, 16 spheroids per μL in 50 μL domes.

- In experiments with re-exposure to inflammation, was the inflammatory cocktail only added for 16 h prior to differentiation or did the treatment continue throughout differentiation as well?

Response: The organoids were treated with the cocktail only prior to differentiation. The differentiation medium had no inflammatory cocktail. We have clarified this in the legend and methods (Line 524-525).

- P30 line 703: "Three spheroids per group were differentiated..." – perhaps it was three domes or wells of spheroids per group?

Response: We thank the reviewer for spotting the error. We meant three wells and corrected this in the manuscript. Now Line 775

- The inflammatory cocktail was modified from the original study by Arnauts et al to have lower concentration of TNFalpha. It would be helpful to have the reason for this modification clarified.

Response: In our hands, we observed ~50% colonoid death during differentiation using the cocktail doses published by Arnauts et al, which made metabolic flux analysis technically challenging. Therefore, we modified the protocol to use lower doses that were less cytotoxic. (Line 148-150)

- P19 line 442: please confirm that spheroid expansion was carried out in pure (not 50%) L-WRN conditioned medium with no supplements (i.e. nicotinamide, B-27, etc.)

Response: We thank you for the comment. We have clarified this in the methods (now Line 491 – 492). We used 50% L-WRN conditioned medium as described in the referenced papers (ref 50 and 51 in the manuscript).

Response to Reviewers—References

1. Dulari Jayawardena et al. "Loss of SLC26A3 Results in Colonic Mucosal Immune Dysregulation via Epithelial-Immune Cell Crosstalk". *Cellular and Molecular Gastroenterology and Hepatology* 15 (2023): 903-919
2. Rial, Eduardo, et al. "Lipotoxicity, fatty acid uncoupling and mitochondrial carrier function." *Biochimica et Biophysica Acta (BBA)-Bioenergetics* 1797.6-7 (2010): 800-806.
3. Cortez-Pinto, Helena, et al. "Lipids up-regulate uncoupling protein 2 expression in rat hepatocytes." *Gastroenterology* 116.5 (1999): 1184-1193.
4. Armstrong, Michael B., and Howard C. Towle. "Polyunsaturated fatty acids stimulate hepatic UCP-2 expression via a PPAR α -mediated pathway." *American Journal of Physiology-Endocrinology and Metabolism* 281.6 (2001): E1197-E1204.

RESPONSE TO REVIEWERS

We thank the reviewers for their feedback, which has significantly improved the quality of our manuscript. In response, we have clarified the images pointed out by the reviewers. Below, we provide a point-by-point response to each comment.

Reviewer #1 (Remarks to the Author):

The authors have adequately addressed my previous concerns and have provided an exciting study

Response: We thank the reviewer for the time and effort.

Reviewer #2 (Remarks to the Author):

The authors addressed all of my concerns. There are some minor errors in the manuscript that should be fixed:

Figure 2: a fragment of transwell is visible at the bottom of panel G

Supplementary Fig. 4:

- Panel E overlapping with a schematic showing sulfite addition prior to differentiation – perhaps this is misplaced
- Panel J shows two identical Basal Respiration box plots overlapping each other

Response: We appreciate the reviewer for their time and effort, and for spotting these errors. We have fixed the images in Fig. 2G, Supplementary Fig. 4E and 4J